# Structure of the Secretory Compartments in Goblet Cells in the Colon and Small Intestine

**DOI:** 10.3390/cells14151185

**Published:** 2025-07-31

**Authors:** Alexander A. Mironov, Irina S. Sesorova, Pavel S. Vavilov, Roberto Longoni, Paola Briata, Roberto Gherzi, Galina V. Beznoussenko

**Affiliations:** 1Department of Cell Biology, IFOM ETS—The AIRC Institute of Molecular Oncology, Via Adamello, 16, 20139 Milan, Italy; 2Department of Anatomy, Ivanovo State Medical University, 153012 Ivanovo, Russia; irina-s3@yandex.ru (I.S.S.); vavilov-007@mail.ru (P.S.V.); 3Department of Energy, Politecnico di Milano, Via Lambruschini 4a, 1, 20156 Milan, Italy; roby.longoni95@gmail.com; 4Independent Researcher, 8355 Station Village Ln., San Diego, CA 92108, USA; pbriata@gmail.com; 5Development Aging and Regeneration Program, Sanford Burnham Prebys, 10901 North Torrey Pines Road, La Jolla, CA 92037, USA; roberto.gherzi@icloud.com

**Keywords:** goblet cell, Golgi complex, ER exit site, diffusion model, intracellular transport, kiss-and-run model, secretory granule, Mucin, regulation secretion, multilamellar organelle

## Abstract

The Golgi of goblet cells represents a specialized machine for mucin glycosylation. This process occurs in a specialized form of the secretory pathway, which remains poorly examined. Here, using high-resolution three-dimensional electron microscopy (EM), EM tomography, serial block face scanning EM (SBF-SEM) and immune EM we analyzed the secretory pathway in goblet cells and revealed that COPII-coated buds on the endoplasmic reticulum (ER) are extremely rare. The ERES vesicles with dimensions typical for the COPII-dependent vesicles were not found. The Golgi is formed by a single cisterna organized in a spiral with characteristics of the cycloid surface. This ribbon has a shape of a cup with irregular perforations. The Golgi cup is filled with secretory granules (SGs) containing glycosylated mucins. Their diameter is close to 1 µm. The cup is connected with ER exit sites (ERESs) with temporal bead-like connections, which are observed mostly near the craters observed at the externally located cis surface of the cup. The craters represent conus-like cavities formed by aligned holes of gradually decreasing diameters through the first three Golgi cisternae. These craters are localized directly opposite the ERES. Clusters of the 52 nm vesicles are visible between Golgi cisternae and between SGs. The accumulation of mucin, started in the fourth cisternal layer, induces distensions of the cisternal lumen. The thickness of these distensions gradually increases in size through the next cisternal layers. The spherical distensions are observed at the edges of the Golgi cup, where they fuse with SGs and detach from the cisternae. After the fusion of SGs located just below the apical plasma membrane (APM) with APM, mucus is secreted. The content of this SG becomes less osmiophilic and the excessive surface area of the APM is formed. This membrane is eliminated through the detachment of bubbles filled with another SG and surrounded with a double membrane or by collapse of the empty SG and transformation of the double membrane lacking a visible lumen into multilayered organelles, which move to the cell basis and are secreted into the intercellular space where the processes of dendritic cells are localized. These data are evaluated from the point of view of existing models of intracellular transport.

## 1. Introduction

Glycans are a key factor that shapes intestinal colonization. Breaches and structural weakening of mucus layer can initiate ulcerative colitis [1,2], misunderstood as “goblet cell depletion,” the loss of sentinel goblet cells at the crypt opening [3,4,5,6,7]. Mucins are biosynthesized, glycosylated, and stored in the Golgi of goblet cells. This Golgi represents a specialized machine for mucin glycosylation [7]. Therefore, an examination of goblet cells with high resolution is extremely important [8,9,10,11,12].

The structure of goblet cells was examined in many papers [13,14,15,16,17,18,19,20,21,22,23,24,25,26,27,28]. However, the applied resolution was not sufficient for an analysis of the extremely complicated Golgi. In order to prove the membrane continuity of cisternae, a very high (up to 3 nm) resolution of three-dimensional electron microscopy (3DEM) is required. Only the electron microscopic (EM) tomography could provide this resolution.

Goblet cells are present in both the small and large intestines, located between absorptive enterocytes. The ultrastructure of these cells is quite unique. The basal part of goblet cells is narrower than the apical part, with is filled with secretory granules (SGs) containing highly glycosylated mucins. Their apical plasma membrane (APM) contains short microvilli [27,29,30]. Dendritic cells or other mononuclear phagocytes are frequently observed near the basis of goblet cells [11,30].

The endoplasmic reticulum (ER) surrounds both the nucleus and the Golgi apparatus, while mitochondria are also frequently located around the Golgi. Most mitochondria are concentrated at the basal part of the goblet cells, often beneath the nucleus. The upper part containing of the secretory granule aggregate is not surrounded by the ER or Golgi but is instead is in close proximity to the apical plasma membrane (APM) [30]. In goblet cells where the APM remains intact and mucus secretion into the intestinal lumen is not actively occurring, the APM maintains its structure with short microvilli [27,29,30]. Typically, the number of Golgi cisternae in goblet cells is equal to 7–8 [31], but this number may increase up to 8 in the case of pathology.

Autoradiographic studies have demonstrated a sequential labeling pattern within the Golgi stack: initially, only the first three cisternae on the *cis*-side were labeled with radioactive glucose (Glucose-H^3^), followed by a shift in labeling to the last 4–5 Golgi cisternae, whereas the first three cisternae lose the signal. Then, labeling appears in secretory granules (SGs) located near the Golgi and finally in the subapical-located SGs [27]. Within four hours of injected radiolabeled fucose, the Golgi area becomes unlabeled, while labeling is observed in the apical SGs [32].

Goblet cells produce significant amounts of mucus, with polysaccharide chains in SGs occupying about 75% of the cytoplasmic volume [13,27,33,34]. During their migration along the intestinal villi, goblet cells completely renew their mucus-containing SGs twice [15]. Mucus accumulation is a relatively slow process [30,35]. The size of SGs is variable, and they are capable of undergoing homotypic fusion [36,37,38]. SGs located apically are generally larger. Contacts between cisternal distensions and SGs frequently display membrane junctions consisting of only three osmiophilic dark layers instead of the usual four such layers [30].

Ellinger and Pavelka [39,40] described a consecutive replacement of the lectin labeling within the Golgi cup of goblet cells, corresponding to the process of glycosylation. Initially, labeling was located in the external part of the “Golgi cup”, followed by a signal in the inner part, and finally in the SGs. Selective staining of Golgi cisternae has been demonstrated using peanut lectin conjugate with horseradish; however, SGs do not exhibit labeling with this marker [41,42]. In contrast, the labeling of sialic acid is observed in the last four Golgi cisternae and SGs [40,43]. Mannosidase II (ManII) immunolabeling is exclusively localized in the last (close to the *trans*-side of stacks) *medial*-Golgi cisternae, as well as in the lumen of SGs [44]. Furthermore, an analysis of the localization of small GTPases involved in intracellular trafficking revealed that Rab3D, a marker of regulated secretion, shows a low level of colocalization with the *trans*-Golgi network (TGN) marker TGN38. However, partial colocalization of Rab3D is observed with β-COP (a component of coatomer), and GM130 (a *cis*-Golgi marker) [45].

According to video presented by Gustafsson et al. [11], the Golgi has the shape of two semicylindrical structures containing numerous irregularly shaped holes. Koga and Ushiki [36] described the Golgi as cylindrical or cup-shaped, measuring approximately 8–10 µm along its long axis and about 3–4 µm along its short axis. They described the form of the Golgi ribbon: one as a continuous cylinder and the other as a concave cup with a large central hole above the nuclear region. The *cis*-most cisterna (CMC) appears as a network of perforated membrane islands interconnected by flat tubules, forming a ring-like structure surrounding Golgi craters, likely organized around the ERES [36].

Mature subapical SGs fuse with the APM, leading to the release of mucins into the intestinal lumen [20,27]. Upon secretion, SGs lose their electron density, as observed in previous ultrastructural studies [46]. The secretion process can be stimulated by cholinergic secretagogues such as histamine and pilocarpine, with induce rapid mucus release from SGs; notably, this process is not inhibited by microtubule-disrupting agents such as colchicine [47,48,49].

It is important to evaluate the transport mechanism involved in intra-Golgi transport in goblet cells [50]. Schemes of the main transport models are presented in Appendix A. The diffusion model (DM) and the kiss-and-run (KARM) [51,52,53,54,55,56,57] explain the transport of proteins through the Golgi in these cells well, where secretion is based on the regulated secretion model [30]. The idea of DM emerged from several works [54,55,56,57,58,59]. DM assumes that the Golgi cisternae are permanently interconnected, at least during the passage of cargo through this organelle [54,59]. The peculiar secretory organization could be important for the consumption of antigens from the lumen of the colon and intestine. Moreover, it remains unclear why manipulations of the same miRNA in two very similar cells produce different results on the Golgi [31].

Here, we tried to use our new knowledge for the explanation of contradictory issues in the field of immunity based on Goblet cells and immunity based on the structure of these cells. We analyze the three-dimensional structure of the Golgi complex in goblet cells in the context of existing transport models [30,60,61].

## 2. Materials and Methods

We used six heterozygous mice, which were used as a control in our previous study [31]. All mice were maintained under pathogen-free conditions, and all experiments involving animals were performed in accordance with the Organismo Preposto al Benessere degli Animali of IRCCS Ospedale Policlinico San Martino. Experimental protocols were approved by national regulators (Authorization # 716/2019-PR). Mice were morning starved for 4 h and then prepared for electron microscopy. Animals were removed from the experiment before the end of anesthesia after opening the chest via the intracranial administration of a saturated solution of potassium chloride at a dose of 1–2 mM/kg. Trained persons sacrificed the rats. Death was confirmed by observing the cessation of heartbeat and respiration and the absence of reflexes, in agreement with international standards (https://www.lal.org.uk, assessed on 25 June 2025). While the animals were under the ether anesthesia, the jejune and colon tissue were removed, processed, embedded, sectioned, and stained.

The following antibodies were used: rabbit polyclonal antibodies against GM130 were a gift from Dr. M.A. De Matteis (Mario Negri Sud Institute, Chieti, Italy). The ManI and ManII polyclonal antibodies were from Dr. K. W. Moremen (University of Georgia, Athens, GA, USA). Antibodies against GalT were a gift from Dr. K. Simons (Max Planck Institute, Dresden, Germany). The rabbit polyclonal antibody against giantin was from ThermoFisher (Milan, Italy; Cat # 22270-1-AP). The rabbit polyclonal antibody against Beta-Galactoside Alpha-2,6-Sialyltransferase 1 (ST6GAL1) was from Abbexa Ltd. (Cambridge, UK; Catalogue No: abx004403; dilution 1/50-1/100). The rabbit antibody against the C-terminus of sialyltransferase (ST*SiaIII) was from Abcam Limited (Cambridge, UK; catalogue number Ab222305). The rabbit anti-beta COP antibody was from Abcam (catalogue number ab2899). The rabbit anti Sialyltransferase Polyclonal Antibody was from AbboMax, Inc. (San Jose, CA, USA; catalogue number 602-900).

Th Rabbit Recombinant Monoclonal antibody against Mucin 5AC antibody [EPR16904] was from Abcam Limited (catalogue number ab198294). The Rabbit Recombinant Monoclonal MUC1 antibody [SM3] was from Abcam (catalogue number ab245695). The Anti-Mucin MUC5AC Antibody, clone CLH2 was from Sigma-Aldrich (Burlington, MA, USA; Catalogue number MAB2011). The Polyclonal Antibody to Mucin 5 Subtype AC (MUC5AC) was from CLOUD-CLONE CORP (23603 W. Fernhurst Dr., Unit 2201, Katy, TX, USA). Rabbit polyclonal α-LC3 (EM dilution 1:100) was from Abgent (San Diego, CA, USA; catalogue number AP1800a-ev).

Nanogold-conjugated Fab fragments of anti–rabbit IgG and Gold Enhancer were from Nanoprobes (Yaphank, NY, USA). Protein A conjugated with colloidal gold was from Dr. J. Slot (Utrecht University, Utrecht, The Netherlands). Anti–rabbit, anti–mouse, and anti–sheep antibodies conjugated with Alexa 488, Alexa 546, and Alexa 633 were from Molecular Probes. Unless otherwise noted, all other chemicals and reagents were obtained from previous sources [59,62,63,64] or from Sigma-Aldrich or ThermoFisher.

Electron microscopy (EM) analysis of samples was performed as described previously [65,66]. The abdomen cavity was opened, and proximal part of the large intestine was cut with the fresh razor blade, and samples were immediately placed into the fixative composed of 2.5% formaldehyde and 2.5% glutaraldehyde in 0.1 M sodium-cacodylate buffer (pH 7.4) (from Electron Microscopy Sciences company, Hatfield, PA, USA). Then, samples were washed with the cacodylate buffer. After washing, the samples were postfixed in 1% OsO_4_ containing K_4_[Fe (CN)_6_] (15 mg/mL) in 0.2 M sodium cacodylate buffer at room temperature for 1 h, followed by 6 times rinsing in 0.1 cacodylate buffer (pH 6.9). Then, the samples were sequentially treated with 0.3% Thiocarbohydrazide in 0.2 M cacodylate buffer for 30 min and 1% OsO_4_ in 0.2 M cacodylate buffer (pH 6.9) for 30 min. Then, samples were rinsed with 0.2 M sodium cacodylate (pH 6.9) buffer until all traces of the yellow osmium fixative had been removed. The samples were subsequently subjected to dehydration in ethanol and embedded in Epoxy resin at RT and polymerized for at least 72 h at 60 °C in an oven. Immune EM based on cryo-sections or nanogold enhancement was performed exactly as it was described [59,63].

### 2.1. Electron Tomography

An ultramicrotome (Leica EM UC7; Leica Microsystem, Viena, Austria) was used to cut 200 nm serial semi-thick sections. These sections were collected on 1% formvar films adhered to slot grids. Both sides of the grids were labeled with fiduciary 10 nm gold (PAG10, CMC, Utrecht, The Netherlands). Tilt series were collected from the samples from ±65° with 1° increments at 200 kV using Tecnai20 electron microscopes (FEI, now ThermoFisher Scientific, Eindhoven, The Netherlands). Tilt series were acquired at a magnification of 7800×, 9600×, 11,500×, 14,500×, or 19,000× using software supplied with the instrument. The nominal resolution in our tomograms was 3 nm, based upon the section thickness, the number of tilts, tilt increments, and tilt angle range. The IMOD package and its newest viewer, 3D IMOD 4.0.11, were used to construct individual tomograms. Tomograms were calculated, analyzed, and segmented using the IMOD software package as it was described by Beznoussenko et al. [65,66]. We examined only the absorptive enterocytes covering the tips of rat intestine villi. The ER exit sites were identified according to the features proposed by Bannykh et al. [67]. The trans-pole of the stack and its resting state of the Golgi were identified according to Mironov et al. [68]. Dendritic cells were identified on the basis of ultrastructural criteria by Rescigno et al. [69,70,71]. The COPII-coat was identified based on the following criteria: the presence of a less osmophilic band between the membrane and the coating. In contrast, the COPI-coat appears to merge almost completely with the membrane.

### 2.2. Stereology

For each time point, we statistically examined 5–6 pairs of samples [72,73]. Each pair was composed of one control and one experimental animal. These two samples were subjected to processing together and were embedded in one block, which was sectioned as a united sample. In the control and in the experimental sample, we randomly selected three cells. The upper edge of the Golgi cup was determined based on the following feature: above this edge, the ER cisternae disappeared and secretory granules are close to the basolateral plasmalemma. Using 3 random slices, we calculated these parameters. An average value was calculated for each animal and used as a statistical unit. The main part of the quantitative analysis was performed on goblet cells of the colon.

We have placed the test grid on random sections of Golgi cisternae and counted the intersections (I) between the test-line and membranes of cisternal rims and membranes of flat zones of cisternae. The ratio between the surface area of rims and surface area of cisternae was estimated. A morphometric grid was applied, and the number of intersections of the grid lines with the membranes of all cisternae (I_cist_), flat parts of cisterns (I_flat_), membranes of vesicles with a diameter of 48–59 nm (I_ves_), and the membranes of the cisterna rims (starting from the end of the flat part of the membrane [I_rim_]) was counted, with membranes of secretory granules (I_sg_). At the same time, we calculated the number of grid points that fell on the Golgi ribbon (P_gr_) and on the secretory granules slice (P_sg_). Next, we calculated the ratio of I_rim_/I_flat_; I_sg_/I_cist_; I_ves_/I_cist_; [I_sg_/P_sg_ ]/[(I_cist_ + I_ves_)/P_rp_/(I_sg_/P_sg_]. For the identification of COPI- and COPII-coats we used the following criteria. COPII-coated buds have a diameter equal to 70–80 nm, and their coat is 1.1-fold thicker. COPI-coated buds have a diameter equal to 50–55 nm, and their coat is more osmiophilic than the COPII-coat [68,74,75].

On the cryosection labeled with antibodies, we performed all necessary controls and measured the ratio of the labeling density (LD) over the organelle and the background, for which LD over the inner mitochondrial membrane (IMM) as the analogue of the background was used. Then, the first LD was divided into the second. Next, the same LD was calculated for the neighboring enterocyte and was considered as 100 percent. The first LD was divided into the second one. Then, the ratio expressed in percents was calculated.

Then, the mean value was used as a statistical unit for statistical analysis. Data were analyzed with Mann–Whitney tests. Cumulative probability distributions were compared using the Kolmogorov–Smirnov test. Data are reported as the mean ± SD. Statistical values can be found in the figures and figure legends. The statistical calculation was performed with GraphPad Prism version 7 and SigmaPlot version 12.5 software. A difference was considered as statistically significant when *p* < 0.05. Statistical analyses were performed using GraphPad Prism software version 9.0 for MacOS (GraphPad Software) as described [31,59,76].

## 3. Results

Using the volume EM (vEM) and immuno-EM (iEM) based on cryosections, we analyzed more than 4000 transmission EM (TEM) and serial block-face scanning electron microscopy (SBF-SEM) images and 85 EM tomograms. Only goblet cells located at the tip of the jejunal villi and on the luminal surface of the colon were examined. Goblet cells within the crypts were not assessed.

### 3.1. Structure of the Golgi

The structure of the Golgi apparatus in goblet cells is similar to that described in the literature (see Section 1). No significant difference was observed in the Golgi apparatus per se of goblet cells between the jejunum and the colon. Figure 1 demonstrates representative examples of images made with the routine TEM sections (Figure 1I–Q); the SBF-SEM images (Figure 1A–H) and EM tomography (Figure 1M) are presented. On vertical sections, the Golgi has a horseshoe-like appearance (Figure 1N,Q), while on horizontal sections, it appears circular. Non-compact zones were not found. Furthermore, the bottom of the Golgi cup is not solid; large holes are present and surrounded by cisternal rims (Figure 1A–D,I–H,M–Q). The shape of the Golgi is similar to a cup. Dendritic cell protrusions (or dendritic cells themselves) surround the base of most goblet cells (Figure 1A–D). Also, the Golgi cup is shown in Figure 2A,D, as presented by Mironov and Beznoussenko [30].

The Golgi stacks are long (Figure 1A–K,M–Q) and consist of 5–10 cisternae (with their average equal to 7–8 cisternae; Table 1). The scheme of the Golgi cup is presented in Figure 2E,F. Appendix A demonstrate different shapes of the Golgi spiral based on the mathematical model explaining the formation of the spiral generated by a helicoid surface. The schemes of Golgi stacks with tangential connections between rims and with vertical connections between rims are illustrated by Appendix A.

We performed a thorough manual analysis of the full data set of serial images of six goblet-shaped cells using serial block-face scanning EM (SBF-SEM). To check the hypothetical presence of a free cisternal island during three-dimensional reconstruction of the Golgi ribbon, one would expect to observe a specific sequence in the serial images: first, a break in the continuous dark line representing the Golgi stack; next, the appearance of a new, isolated dark line, completely separated from both ends of the original structure; and finally, the disappearance of this isolated line while the break remains. However, we did not observe such a sequence in any of the analyzed data sets. Appendix A shows a scheme explaining our strategy directed to find the isolated Golgi stack.

Edges/rims of the Golgi cup are shown in Figure 2A,B,D. Vertical sections of resting (non-secreting) goblet cells, with the apical plasmalemma, which is not fused with subapical granules, are shown in Figure 1B,E,F and Figure 2F,G presented by Mironov and Beznoussenko [30].

In the goblet cells examined, approximately 15% of the Golgi structure exhibited a cylindrical shape, 65% appeared as a cup with a large hole in the bottom, and 20% resembled a cup with small bottom hole(s). In some cases, through-holes were observed in the walls of spiral-, cone-, or cylinder-like stacks (Appendix A). The tip of the cone is located directly about the nucleus and corresponds to the center of the spiral’s rotation. This region includes a tangential zone and contains the center of the spiral’s rotation and a tangential part with narrow pores. A specific point of contact, located at the base of the Golgi structure, appears to serve as a reference point for the formation of the Golgi spiral.

Tomographic images (Figure 2A–D,I) and three-dimensional reconstructions (Figure 2G–K) of the Golgi support the conclusion derived from routine transmission EM sections. The first three from the cis-side externally localized osmiophilic cisterna are visible in Figure 2B. The rims of the Golgi contain a low number of COPI-coated buds (Figure 2A,B,D). Fenestrae in the cis-most cisternae are shown in Figure 2C.

### 3.2. The Structure of the ER–Golgi Interface

The outer surface of the Golgi stack is surrounded by ER cisternae (Figure 2A,B) that contain ERESs, in contrast to absorptive enterocytes, where ERESs were not found [74,75]. The ERESs are located opposite the craters visible on the outer surface of the Golgi stack (Figure 2G,H). ERESs are relatively rare near the edges of the “Golgi cup” and around the rims of the large holes within it. ERESs were identified by presence of COPII-coated buds on the ER membrane (Figure 2A,B,D,I–K), in accordance with the criteria established by Bannykh et al. [67].

We observed only two COPII-coated buds and did not find 70–85 nm vesicles in the ERES zones. Also, in our samples, we observed rare connections between ERESs and the rims of Golgi cisternae (Figure 3A–G,I,J). ER–Golgi connections were mostly observed within the craters and typically appeared as a varicose (beaded) tubular structure, with could be subdivided into two distinct parts. The ER were most commonly connected with the rims of the first three osmiophilic cisternal layers of the Golgi ribbon, usually within craters regions (Figure 3A–D). These observations suggest that craters serve as the main entry sites to the Golgi apparatus and ERESs associated with Golgi cisternae through varicose tubular connections. The computer-generated three-dimensional model of these bead-shaped ER–Golgi connections within craters is shown in Figure 3C,D. Additionally, the plasticine 3D-model illustrating similar connections within craters is presented in Figure 3E,F.

The outer surface of this Golgi cup is partially covered with the fenestrated CMC (Figure 2C). The lumen of the first three Golgi cisternae (or cisternal layers) starting from the external surface of the Golgi cup are more osmophilic than those of the 4th–7th cisternae (Figure 2B,D). These craters represent cavities where the first three cisternae have the pores. The diameter of these pores decreases progressively toward the fourth cisterna layers. These pores are vertically aligned, creating a columnar arrangement. The hole in the outermost cisternal layer (the outer layer of Golgi stack or spiral) is largest, with a diameter of approximately 500–600 nm. The hole in second layer is smaller, and the third usually has the smallest diameter (about 150–200 nm). In most cases, the fourth cisterna is visible on the bottom of the crater. The craters may reach as deep as the fifth cisternae, but this occurs rarely. Craters are filled with round membrane profiles and varicose tubules that connect the ER to one of the cisternae forming the crater (Figure 2G,H and Figure 3A–D). The crater is visible in Figure 4C,D, presented by Mironov and Beznoussenko [30]. Also, our three-dimensional analysis reveals the presence of ER–Golgi connections (Figure 3A–N).

### 3.3. Formation of Cisternal Distensions

In the fourth cisterna, mucin deposition begins, accompanied by luminal expansion. These distensions appear to be displaced toward the distal end of the single cisterna within the cylindrical or conical Golgi structure. Clusters of the 52 nm vesicles are observed between Golgi cisternae at different levels of the Golgi ribbon and between secretory granules (Figure 4 and Figure 5D). Accumulations of COPI-dependent vesicles between cisternae on the *trans*-side and SGs may serve to increase the surface area of SGs during the beginning of mucus secretion (Figure 4H–M).

Narrow pores are frequently observed near the junctions where mucin-filled cisternal distensions connect to the other parts of the same cisterna (Figure 5A–C,E,G). Notably, 52 nm vesicles fusing with the cisternal membrane are often found in close proximity. We also observed relatively long membrane contacts between the membranes of 52 nm vesicles and the cisterna (Figure 5A–H), suggesting that these sites lack intervening cytosol domains and are primed for imminent fusion [77]. The fusion of these highly curved 52 nm vesicles with the cisternal membrane likely transfers membrane curvature to these places, thereby promoting pore expansion [65].

The space between all seven *medial*-Golgi cisternae is extremely uniform and narrow. The typical *trans*-most cisterna (TMC) was not found. Furthermore, clathrin-coated buds on the membranes of last Golgi cisternae just before their transformation into SGs are rare. Similarly, clathrin-coated buds are not observed on the membrane of SGs and endosomes. Endosomes themselves were rare, and only a few multivesicular bodies were seen in proximity to the “Golgi cup”.

Beginning with the fourth cisterna of the Golgi stack, the thickness of cisternal lumen began to increase, and a precipitate similar in structure to the contents of secretory granules begins to accumulate (Figure 4A). This precipitation of mucins initially occurs in the central areas of the fourth layer of Golgi stack, triggering the formation of cisternal distensions. The thickness of these central distensions progressively increases in the next layers of the stack. Moving toward the distal part of the Golgi ribbon, such distensions occur more frequently and finally reach the size of SGs. Some distensions are localized near the cisternal rims. It seems that due to the high mobility of the Golgi, mucins containing precipitates are moved to the rims. Finally, these distensions become roundish and are concentrated near the rims of the Golgi cisternae. They are separated from the rest of the cisterna by a series of narrow pores, previously described by Beznoussenko et al. [63].

### 3.4. The Problem of Cisternal Rims and Membrane Contacts

The rims of the cisternae are found on the upper edge of the “Golgi glass” and around windows in its craters. Using a stereological approach, we measured the ratio of the surface areas of the rims of the cisternae to the area of their flat zones for the 1st–5th cisternae, whereas for the 6th and 7th and sometimes the 8th cisternae, we did not perform this, since there are no craters in these zones. In addition, in these cisterna layers, we noticed cisterna extensions that change the proportions. This ratio is unusually small (Table 1). In the fifth cisterna, this ratio was equal to 0.19 ± 0.08. However, the formation of distensions in this and more distal cisternal layer make calculations rather difficult.

Our stereometric estimations demonstrated that when the width of cisterna is equal to the cisterna is 1 µm, the ratio of the rim area to the area of the flat part of the cisterna membrane is 0.08. If the width of the cisterna is equal to 2 µm, then the ratio is 0.04, and if we take a spiral with one edge and width, cisternae are 3 microns, then the ratio is 0.012. A comparison of our number with those obtained during modeling suggests that most likely the single wide Golgi cisternae forms helicoidal spiral with holes (see Appendix B). A similar structure is found in the ER in cisterna packages, where all cisternae are connected by means of membrane tubes in the center of the stack [78]. Evidence for the fusion of SGs is presented in Figure 3O,P. Membranes of Golgi cisternae and SGs have zones where membranes are tightly attached to each other and form a contact composed of five layers (or three dark layers; Figure 4A–I). Vesicles with a diameter of 52 nm tightly attached to the cisternal membrane are frequently observed near these pores. Some vesicles were connected (Figure 4J–M).

These close membrane contacts represent the sign of future fusion between these membranes [77]. Additionally, localized tight attachments between adjacent cisternae were observed. These contacts between the cisternae are quite stable in width, although in rare cases, these contacts converge sharply, forming a structure composed of five membrane layers. This type of attachment is characterized by the presence of three dark lines of membranes instead of the usual four, suggesting a special form of membrane contacts (Figure 4A–I). Such contacts, including those between secretory granules, are illustrated in Figure 3G,H, presented by Mironov and Beznoussenko [30].

Vesicles tightly attached to cisternal membranes are presented in Figure 5C,E–H. The pores in cisternae are shown in Figure 5A–C,E,G. Also, a narrow cisternal pore located near cisternal distensions is shown in Figure 3C,D presented by Mironov and Beznoussenko [30]. Additionally, vesicles in close contact with the cisternal membranes, indicative of potential membrane fusion, are also shown in Figure 3A–D, presented by Mironov and Beznoussenko [30].

### 3.5. Immune Electron Microscopic Analysis

Next, using Takayasu cryosections, we examined the distribution of GM130, ß-COP, and giantin in the Golgi of goblet cells and compared it with that in the neighboring enterocytes. GM130 was localized along the external surface of the Golgi ribbon, whereas ß-COP was mostly present on the cisternal rims. Giantin was observed within stacks in the enterocytes. Notable, the expression levels of ß-COP and giantin in the Golgi in goblet cells were significantly lower compared to those observed in the Golgi of the neighboring enterocytes with microvilli.

Further, in order to check whether the 52 nm vesicles could be carriers, we performed immune EM labeling for routinely used and important goblet cell [39,40] and Golgi enzymes, namely, GalT (Figure 6G,I) and SialTF (Figure 6H). No gold over the 52 nm vesicles was visible. On the other hand, trying to understand our previous observations [31], we also labeled the Golgi in goblet cells and neighboring enterocytes for giantin (Figure 6A), ß-COP (Figure 6B,D–F), and GM130 (Figure 6C; Table 2). The expression of these markers in the Golgi of goblet cells is lower than in the Golgi of absorptive enterocytes.

The concentration of these enzymes in 52 nm vesicles is significantly lower than in cisternae. Labeling for Muc1 and LC3 was detected over MLOs. Labeling for LC3 and Muc1 over MLOs was 4.1 ± 0.9-fold and 4.6 ± 1.1 higher than the background. The results of the cryo-immune EM analysis are shown in Figure 6A–F and Table 2 (Appendix A).

### 3.6. Structure of Secretory Granules

Inside the Golgi cup, above the Golgi cone, or on the flat Golgi spiral, a mass of SGs was observed (Figure 1A). These SGs form a substantial mass with only rare membrane inclusions. An increase in the number of cisternal distensions was observed on the *trans*-side of the Golgi stack. Some SGs fuse with each other. An example of fusion of SGs is shown in Figure 3J,K.

In goblet cells where the APM with short microvilli is ruptured, SGs with less dense contents are more frequently observed. These cells also exhibit an increased surface area, which becomes sequestered into multi-lamellar organelles (MLOs) that may later contribute to the formation of new compartments with the secretory pathway.

The membranes of SGs are tightly attached to each other and, in some cases, almost fused, forming a structure composed of five distinct layers instead of the conventional seven. Within these zone of close membrane contact, the membranes are arranged as three osmiophilic (electron-dense) layers separated by two lighter (less dense) layers. Representative contacts between cisternal distension and SGs, as well as between adjacent SGs, are illustrated in Figure 4A–I.

We observed narrow pores near the boundary cisternal distensions and the 52 nm vesicles (yellow arrows) attached to membrane of this cisterna near pores (Figure 5A–H). Also, we found the 42 nm vesicles (Figure 5I) previously demonstrated in absorptive enterocytes [76]. Their role is unknown.

The condition of SGs and APM represents the ultrastructural difference between resting and secreting goblet cells. In the cells where there is no fusion of APM with SGs, SGs are spherical and smaller than in cells where mucus is released. In contrast, secretion is associated with the fusion of the APM with multiple subapical SGs, leading to the formation of large, elongated SGs. There, the length of the elongated SGs could reach 5 µm (Figure 1F–H; Table 1). Notably, membrane contacts characterized by tree dark lines were observed in 21% of cells with an intact APM and only in 5% of cells undergoing secretion.

In order to check whether the 52 nm vesicles could be carriers, we performed immune EM labeling for routinely used Golgi enzymes, namely, GalT (Figure 6G,I) and SialTF (Figure 6H). No gold over the 52 nm vesicles was visible. On the other hand, trying to understand our previous observations [31], we also labeled the Golgi in goblet cells and neighboring enterocytes for giantin (Figure 6A), ß-COP (Figure 6B,D–F), and GM130 (Figure 6C; Table 2). Expression of these markers in the Golgi of goblet cells is lower than in the Golgi of absorptive enterocytes.

### 3.7. Turnover of Membranes of Secretory Granules

After the fusion of a large, elongated SGs with the APM, the surface area of the APM increases dramatically. However, the fate of this excess membrane remains unclear. Therefore, using EM tomography, we examined apical portions of goblet cells. In one type of goblet cell, we observed the formation of APM protrusions with SG inside. As a result, SG was surrounded with a double membrane (Figure 7A,B,J,K). In the apical parts of other cells, we found MLOs connected by a double membrane to the APM (Figure 7L–O and Appendix A). On the other hand, we detected MLOs connected to the outer mitochondrial membrane (Appendix A. Also, we observed the accumulation of numerous MLOs in the ER (Figure 8A,C,D) and the Golgi (Figure 7E–I). Additionally, we detected mitochondria where MLOs was connected with the OMM (Figure 7H,I). Finally, MLOs are visible in the space between the BLPM of epithelial cells (Figure 7E–I and Figure 8B,E–J). Images of the MLO secretion were visible (Figure 8E,H). The presence of MLOs at different levels of secretory pathway in goblet cells is visible in Figure 4A,B,E–H presented by Mironov and Beznoussenko [30]. MLOs were positive for labeling for LC3 and Muc1 (cryosections; Appendix A and nano-gold technology; Appendix A; Table 2). This suggests the role of autophagy in the process of APM membrane elimination. Also, these data demonstrate an alternative model for the uptake of antigens for the intestinal lumen by goblet cells.

## 4. Discussion

In this study, the fine structure of the Golgi in goblet cells was examined using high-resolution 3DEM. Synchronization of the synthesis and maturation of SGs is extremely challenging. For example, in cells where at least one SG has already fused with the APM, the stimulation of secretion may trigger the fusion of only the next subapical secretory granule. Furthermore, the Golgi do not respond synchronously to secretory stimuli, contributing to substantial heterogeneity in their secretory activity [79,80]. For our purposes, we divided goblet cells into two main categories: (1) cells in which at least one SG was observed to have fused with the APM, and (2) cells in which no such fusion was detected. Our ultrastructural analysis did not reveal a significant difference in morphology of the Golgi between cells in the mucin-secretion stage and those with an intact APM. The only consistent distinction was observed in the size and shape of subapical SGs, which tended to be large and more irregular in size in secreting cells.

Taking into consideration that the ratio between the surface area of cisternal rims and the surface area of plane parts of cisternae is extremely low, we conclude that stacks with connections according to [81,82] are rare. In goblet cells, the cisternal rims are visible in the area of craters, at the ends of the cup, and along the perforations in the cup-like Golgi ribbon. The relative low frequency of rim visualization, in comparison with very long, flat parts of Golgi cisternae stacks, supports the hypothesis that the Golgi spiral is organized according to the helicoidal surface (see Table 1 and theoretical predictions in Appendix B). Thus, in these cells, the Golgi could be presented as one wide cisterna in the spiral configuration [83]. Usually, in goblet cells, the Golgi contains of 6–10 layers of the single cisterna. Models illustrating this structural organization are presented in Figure 2E,F and Appendix A.

Our analysis suggests that the Golgi apparatus can form a cup-like structure (see explanation in Appendix A). The beginning of the cisternal spiral ribbon that forms the Golgi “glass” or “cap” is located externally, on the outer surface of the Golgi apparatus. In contrast, the end of the cisternal spiral, forming the layered structure and exit from the Golgi, is positioned internally, within the “Golgi cup”. The “Golgi spiral” is formed according to the helicoidal surface regularities (Appendix B).

We did not find the classical trans-most cisternae (TMC). Endosomal structures are rare in the Golgi. The surface area of Golgi cisternae within the ribbon structure is comparable to that reported in prior studies; however, the overall volume is larger (Table 1) compared to in vitro observations [64,84,85,86,87,88]. This increase in the Golgi size is likely due to the accumulation of mucins within the lumen of cisternae, with increases in their internal volume.

Several structural models of the Golgi ribbon may be considered: (1) a stacked configuration with tangential tubular connections between neighboring cisternae; (2) a stack with vertical tubular connections between adjacent cisternae [68] (see also Appendix A); (3) a flat, spiral-like arrangement where cisternae lie horizontally, surrounding a central hole to form a ribbon-shaped stack, as described by Tanaka [89]. The central holes in this configuration can vary in size. (4) A novel model is proposed here: a spiral Golgi ribbon configured in the form of either a cone (Appendix A; Appendix B) or a cup (Figure 2E,F). We also did not observe completely isolated stacks. Also, Harada et al. [90], using ultrahigh-speed imaging, did not observe permanently existing ministacks—these structures merged immediately after formation. Based on these observations, we hypothesize that the Golgi apparatus forms a spiral composed of a cycloidal single cisterna with larger diameters at the upper holes and smaller diameter t lower one.

In the cup-shaped model, the spiral begins on the outer surface of the Golgi cylinder and terminates internally, with the single, continuous cisterna progressing inward along the spiral path. A helicoidal model in which the ends of each cisternal “disk” layer occur over one another to varying degrees, results in a conical appearance. The greater the overlap of these concentric elements, the sharper the cone shape. This configuration is consistent with mathematical modeling results (Appendix B) and supported by geometrical studies [91,92,93]. An alternative interpretation considers the Golgi ribbon as a single continuous stack arranged into an irregular spiral with either left- or right-handed helicity, as proposed by Terasaki et al. [78].

Judging from the images of COPII-coated buds in samples obtained after the ultra-fast freezing of cells [74,94,95,96], the diameter of the COPII-dependent vesicles should be equal to 70–85 nm. We did not find such vesicles. Thus, goblet cells do not have COPII-dependent vesicles. This observation is similar to the situation of absorptive enterocytes [76]. Not all craters and ERESs have visible connections. This suggests that that such connections are transient. Under conditions where membrane compounds are constantly formed with the help of SNARE s and calcium and then dismantled by COPI-coated buds and BARS, such connections are difficult to capture in samples. In many cases, the connections can be detected only using several types of vEM. We identified four such bead-like connections in our samples using electron tomography (Appendix A).

The osmophilic feature of the first three cisternal layers of the Golgi cup is caused by osmium precipitation induced in turn by the presence of glucose on mucins [97,98,99]. During diffusion along the Golgi ribbon, glucose is trimmed [100,101]. The addition of sialic acid to polysaccharides increases their sensitivity to low pH, resulting in the precipitation of mucins and the formation of distensions [40]. These distensions are moved toward the end of the cisternal spiral, likely stopping the dynamic mobility of the Golgi.

Aggregates of COPI-dependent 52 nm vesicles and rare short bead-like tubules are observed between Golgi cisternae and SGs. Presumably due to the very fast movement of the Golgi [90], the 52 nm vesicles form clusters.

SGs and cisternal distensions form peculiar tight contact sites composed of only three osmiophilic lines. The typical membrane contacts between organelles have four osmiophilic lines, which are formed through the deposition of metallic osmium on the heads of lipids [99]. In these tight contacts, the two inner electron-dense lines converge and practically form one, albeit slightly wider, line. We previously described similar contacts between vesicles and cistern membranes during vesicle fusion [77]. Such tight contacts are likely formed under conditions when membrane proteins are eliminated from the contact zone or when cytosolic domains of membrane proteins are very short. This phenomenon indicates that membrane proteins are either excluded from these contact zones or possess very short cytosolic domains, such as Man-II or certain membrane mucins [102,103,104]. However, Man-II is not completely excluded from these contacts. Indeed, in the goblet cells of both the duodenum and colon, Man II is exclusively localized in the *trans*-Golgi cisternae and within the lumen of SGs. In contrast, the *cis*-cisternal layers are Man-II-negative [44].

When the intracellular concentration of calcium increases, SNAREs are less important for membrane fusion [105]. Membranes forming such kinds of contacts could fuse even without augmentation of the calcium concentration. Indeed, when SGs form these types of tight contacts, they can fuse even without calcium [34,106]. Following fusion of SG remnants of the membrane, contact is often observed within the content of the newly formed SGs [36,107].

According to the vesicular model [79], COPII- and then COPI-dependent vesicles are transport carriers (Appendix A). We proved that at least in yeast, COPII-vesicles do not exist [50,66]. Moreover, COPII-vesicles are not important for the ER–Gogi transport of several cargos, and cells could survive without them [50,108]. As we mentioned above, we found only two COPII-coated buds within ERESs and did not observe the 70–85 nm vesicles in the ERES zones. In 2000, Pepperkok et al. [109] proved that COPI-dependent vesicles exclude both anterograde and retrograde cargos. As we indicated above, these vesicle clusters are not adapted to transport, because they have no interconnecting strings. Also, the surface area of the rims is insufficient to support the use of 52 nm vesicles as transport carriers. In the 52 nm vesicles, the concentration of GalT, SialTF (current report), and nucleotide sugar transporters (NSTs), which form oligomers being too large for the 52 nm vesicles [110,111,112,113], is lower than in cisternae.

We believe that the model of intra-Golgi transport per se corresponds to the DM (Appendix A) because there are no separate stacks and non-compact zones between them. Additionally, taking into consideration the low ratio between the surface area of the rims and the surface area of flat cisternal membranes, we concluded that the Golgi spiral is formed by wide cisterna, although with perforations.

Data by Cutrona et al. [108], Weigel et al. [58], and Kasberg et al. [114] also questioned the existence of COPII-coated vesicles. Two competing models remain, namely the DM and KARM. DM is well-suitable for regulatory secretion mechanisms [30]. The diffusion model (Appendix A) assumes the constant continuity, whereas the KARM (Appendix A) needs breaks between compartments of the secretory pathway. Appendix A demonstrates that the KARM function in the following manner, namely the fusion between a carrier domain and a distal compartment, occurs before subsequent fission of the connections between the cargo domains and the proximal compartments. The bead-like shape of connections observed between the ER and the Golgi suggest that these connections are unstable and would be subjected to fission. ER–Golgi connections have been described several times (reviewed by Mironov and Beznoussenko [79]).

The adoption of diffusion as the primary mode of intra-Golgi transport in goblet cells does not imply complete membrane continuity throughout the secretory pathway. Instead, transient discontinuities exist, particularly between the ERES and the Golgi cisternae (especially in the crater area), between the end of the Golgi spiral and the forming SGs, and between these SGs and the APM. These transient (not permanents) continuities need SNAREs. The SNARE molecular machinery is necessary for the function of the KARM. However, the SNARE proteins responsible for mediating membrane fusion at these junctions remain incompletely characterized [115,116,117]. At least the role of Ykt6 remains unclear [115].

Our findings in goblet cells suggest that diffusion-based intra-Golgi transport may not be unique to these cells but could also apply to other secretory cell types engaged in regulated secretion, where the secretory granule contents condense post-Golgi. This model offers a unifying mechanism for efficient cargo transport along the Golgi apparatus in cells specialized for regulated secretion. Since we did not find isolated stacks and did not find breaks in the Golgi ribbon, it is likely that the mucins simply diffuse along the Golgi spiral. The fact that there are very few breaks between the Golgi compartments is also indicated by the very small ratio of the surface area of the rims to the surface area of the cisterns. However, the fact that there is no osmium deposition in the ER, even in places close to the cis-Golgi, suggests that there is a break in the permanent continuity of membrane structures and the constantly renewed continuity is unidirectional due to the coordination of bead-like tube ruptures. The second break in the permanent continuity of the secretory pathway is at the level of contact between secretory granules and mature rounded extensions of cisterns. Here, there is a mechanism described by us for all cargo-enriched domains [63], namely the distension is separated from the cisternae by pores, which can increase in size after fusion with a 52 nm vesicle. Therefore, we suggest that in goblet cells, the KARM and diffusion model (only at the level of the Golgi) form a combined model. Narrow pores accumulate between the distensions and the adjoining flat cisternal part, where clusters of small vesicles are usually observed. The fusion of these vesicles with the cisternae may enlarge the pores, facilitating the separation of the SGs [65].

Another unclear problem is the following. When mucus secretion begins and the lumen of the distal SGs is emptied, the surface area of the APM increases dramatically. One possible mechanism aimed to eliminate these excessive membranes could be based on the protrusion of the membranes derived from the SG just fused with the APM. This protrusion may incorporate another SG. As a result, a bubble-like structure is formed, in which SG is surrounded by a double membrane. This double-membrane structure eventually detaches from APM, and then, it is subjected to destruction.

A second mechanism could involve the collapse of the SG lumen. After secretion, SGs positive for Muc1 collapse and form a cord-like structure composed of two membranes tightly attached to each other. This cord may gradually twist and condense, eventually transforming into a small MLO. MLOs formed from the outer mitochondrial membrane (OMM) can capture LC3, SNAP29, and STX17 and then fuse with MLOs derived from the membrane of the emptied SG and APM. It is known that MLOs involved in autophagosome formation are derived from the OMM [118,119]. Notably, in goblet cells, proteins involved in autophagosome initiation and elongation are essential for efficient mucus secretion [120,121]. Our hypothesis suggesting the formation of MLO from the APM easily explains the observations reported by Gustafsson et al. [11].

On the other hand, MLOs are rather osmophilic. The accumulation of MLOs along the external surface of the Golgi ribbon, together with the presence of the first three osmiophilic layers of the cis-Golgi cisternae, likely contributed to the reconstruction of the Golgi cup structure visualized in their study. Gustafsson et al. [11] could consider these osmiophilic layers as the TGN filled with the 10 kDa lysine-fixable dextran-biotin.

It is established [6,7,122] that goblet cells can transfer luminal antigens to dendritic cells surrounding goblet cells. According to Gustafsson et al. [11,12], dextran can be taken up by goblet cells from the intestinal lumen, with this phenomenon not observed in neighboring absorptive enterocytes, and could be involved in immunity. However, the mechanism by which dextran might reach the *trans*-Golgi network remains unclear, particularly, which practically disappears in adults [76]. The transcytosis of antigens from the intestinal lumen to the lamina propria needs specific receptors [123]. In adult animals, cationic ferritin attached to the APM is not delivered into the lumen of Golgi cisternae [124]. Also, mucin induced a significant reduction in the transcytosis across epithelial monolayers [28]. These findings suggest that the level of endocytosis in goblet cells is very low. Moreover, we did not observe clathrin-coated buds and membrane invaginations on the APM of goblet cells and also on SGs. This observation is also supported by the ultrastructural data in Figure 1E,F presented in the paper by Mironov and Beznoussenko [30], which demonstrate the lack of clathrin-mediated endocytosis on the APM of goblet cells.

Our model allows us to solve this contradiction. It suggests that the emptied lumen of SGs may become filled with the substances derived from the intestinal lumen. These substances (such as dextran and potential antigens) could attach to the inner surface of SG membranes before their transformation into MLOs, which subsequently traffic toward the Golgi. This process may mimic classical endocytic uptake.

McDole et al. [122] demonstrated that dextran can move from intestinal lumen towards the Golgi apparatus, often surrounding clusters of SGs. We hypothesize that MLOs can also secreted, thereby delivering these antigens to dendritic cells. The secretion of the contents of autophagosomes into the extracellular space through the basolateral plasma membrane is a known phenomenon [125]. Also, we believe that membranes of MLOs could be reused, because the de novo synthesis of fatty acids is rather slow [126,127]. Finally, in goblet cells, the expression level of Golgi-associated proteins such as giantin, COPI, and GM130 are lower than in villous enterocytes. This difference may explain why manipulations involving the same miRNA gave different results in these two cell types [31].

## 5. Conclusions

Three-dimensional analysis of the Golgi apparatus in goblet cells revealed an unusual spiral ribbon-like structure. Several features indicate that the most suitable explanation of our ultrastructural data could be achieved within the frameworks of the DM for intra-Golgi transport within the Golgi per se combined with the kiss-and-run model, as well as the contact zones between the apical plasma membrane and distal SGs. The presence of tight contacts between secretory granules suggests that membrane fusion may be triggered by an elevated intracellular calcium concentration.

## Figures and Tables

**Figure 1 cells-14-01185-f001:**
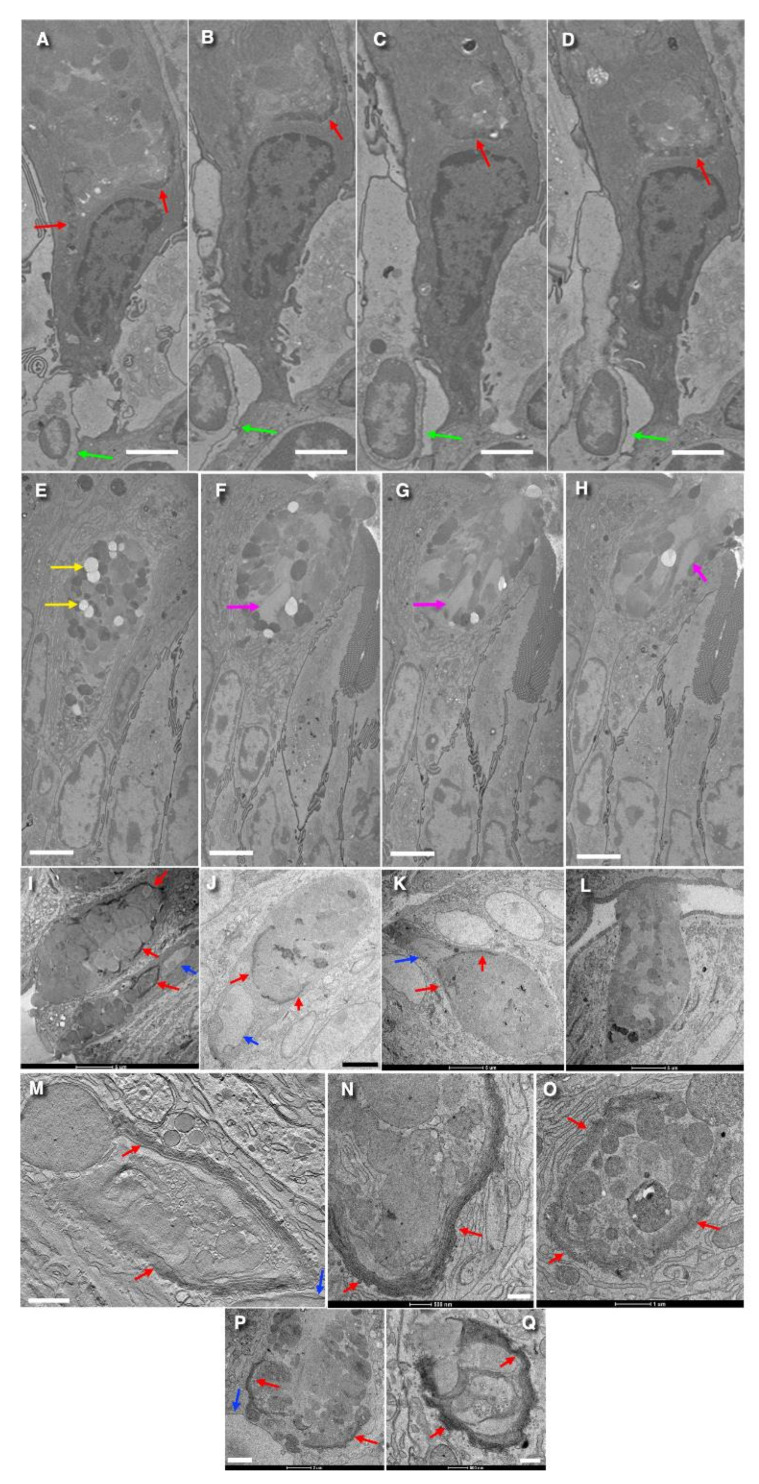
Structure of the Golgi in goblet cells in the jejunum (**A**–**D**,**P**,**Q**) and colon (**E**–**H**,**I**–**O**). Representative examples of images made with the routine TEM sections (**I**–**Q**), 3VIEW or serial block face (**A**–**H**), and EM tomography (**M**). (**A**–**D**) Serial 3VIEW images of goblet cells contacting with dendritic cells (green arrows). Red arrows show different shapes of the Golgi in different vertical sections. (**E**–**H**) Serial 3VIEW images of the goblet cells in the colon containing secretory granules of different types including empty (yellow arrows) and very long ones (purple arrows). (**I**–**Q**) Different shapes of the Golgi cup (red arrows) in different goblet cells. (**L**) Massive secretion of mucin from the Goblet cells. (**J**,**L**) Nuclei (blue arrows) are below the “Golgi cup”. (**M**) Blue arrow (to the right) shows the nucleus below the Golgi cup. (**O**) Secretory granules in resting goblet cells are separated from each other. Red arrows indicate the different shapes of the Golgi apparatus (dark structures in (**A**–**D**,**I**–**K**,**M**–**Q**)). Green arrows show dendritic cells (in (**A**–**D**)). Yellow arrows indicate the “empty” SGs attached to each other (in (**E**)). Purple arrows demonstrate elongated SGs with the length up to 4 µm (in (**F**–**H**)). Blue arrows show the nuclei just below the Golgi cups (in (**I**–**K**,**M**,**P**)). The edge of the Golgi cup (red arrow) with three initial osmiophilic cisternal layers. Green arrows in subfigures (**A**–**D**) show dendritic cells; Purple arrows in subfigures (**F**–**H**) show large SGs; Yellow arrows in subfigures (**E**) show empty SGs; Red arrows in subfigures (**A**–**D**,**I**–**K**,**M**–**Q**) show the Golgi; Blue arrows in subfigures (**I**–**K**,**M**,**P**) show nuclei. Scale bars: 4 µm (**A**–**H**,**I**,**K**,**L**); 5 µm (**J**); 2 µm (**M**,**P**); 500 nm (**N**,**Q**); In Figures (**I**,**K**,**L**,**N**–**Q**), the length of bars is also indicated directly below images.

**Figure 2 cells-14-01185-f002:**
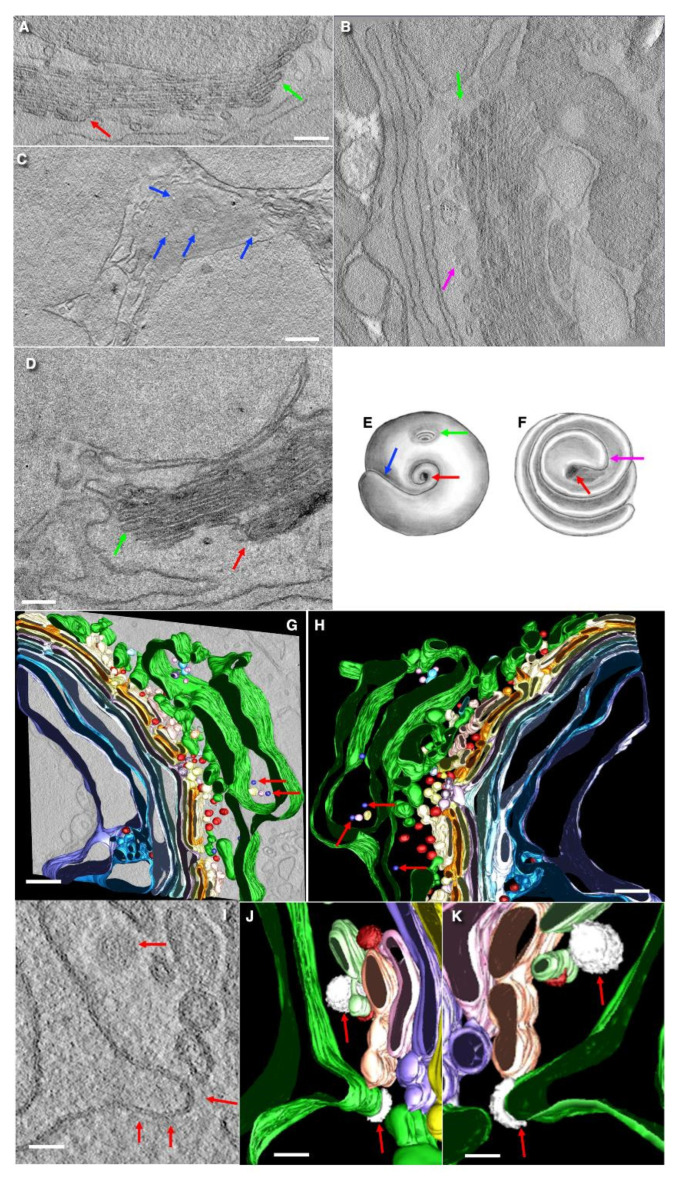
Structure of the Golgi in goblet cells. Tomographic images (**A**–**D**,**I**) and three-dimensional reconstructions (**G**,**H**,**J**,**K**). (**A**,**B**,**D**) Structure of the upper edges (green arrows) of the Golgi cup. Low development of the COPI-coated buds. (**A**) The first cistern (layer) on the external surface is indicated with the red arrow. (**B**) The first three cisternae (green arrow) on the external surface are more osmiophilic than the next Golgi cisternae. Purple arrow indicates the most *cis*-cisterna. (**C**). Tangential section of the Golgi cisternae. Rare narrow pores (blue arrows) are visible. (**D**) The red arrow shows the initial layer of the Golgi spiral. The upper edges of the Golgi spiral cup is indicated with the green arrow. (**E**,**F**) Scheme of the Golgi cup. The blue arrow shows the beginning of the Golgi spiral. The blue arrow shows the beginning of the cisternal spiral. The purple arrow shows the end part of the spiral. Red arrows show the point of rotation (see Appendix A). (**G**,**H**) Three-dimensional reconstruction (3D) of the Golgi ribbon. Green color indicates the ER. Blue color demonstrates cisternal distensions filled with mucins and secretory granules. In (**G**,**H**) blue small spheres (red arrows) near the ER have a diameter of 42 nm. Other spheres near the ER and near blue spheres have diameters equal to 50 nm. (**I**–**K**) COPII-coated buds (white structures) on the ER cisternae. Red color indicated COPI-dependent vesicles. Other colors demonstrate Golgi cisternae. (**I**) Red arrows show COPII-like coat. (**J**,**K**) Red arrows indicate COPII-coat. Structures colored in red indicates COPI-coated bud. Scale bars: 172 nm (**A**,**B**,**D**,**G**,**H**); 275 nm (**C**); 82 nm (**I**–**K**).

**Figure 3 cells-14-01185-f003:**
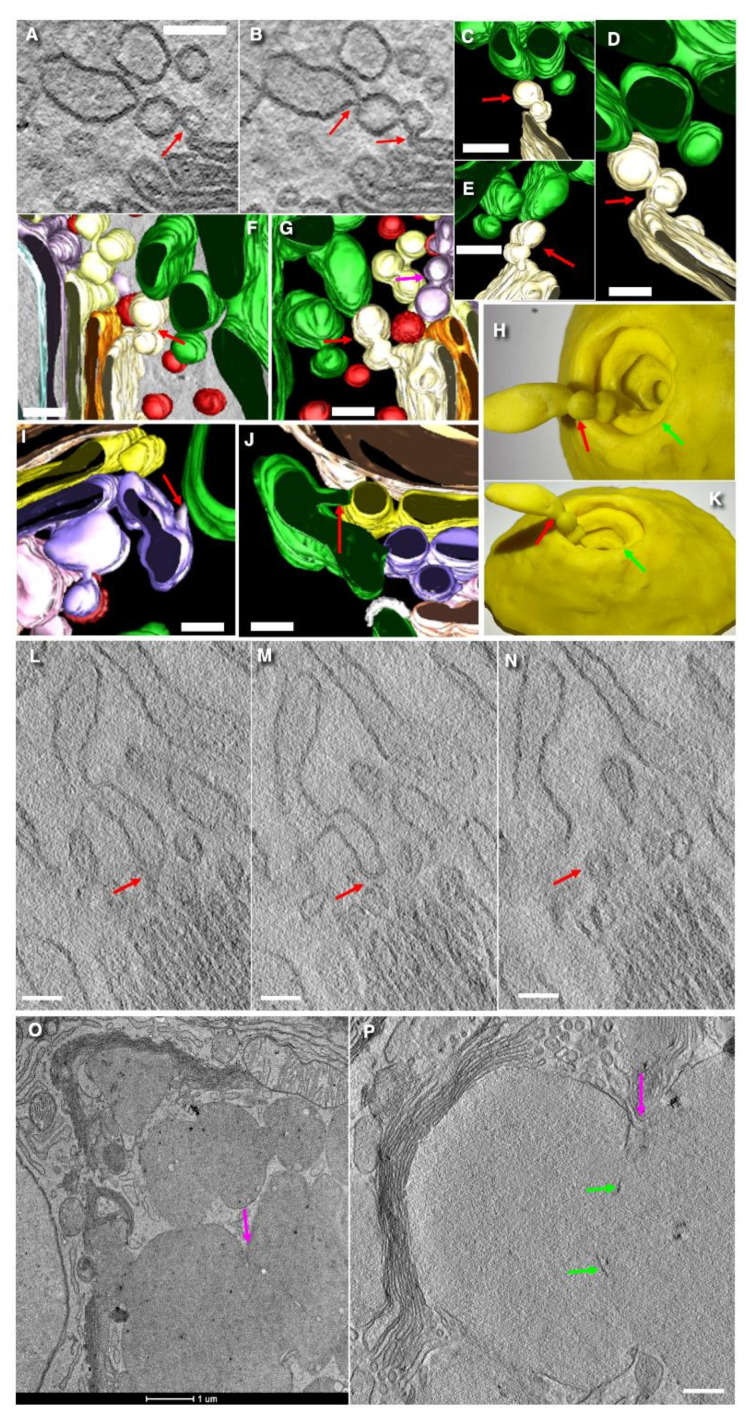
Three-dimensional reconstruction of the outer pole of the Golgi in goblet cells. (**A**–**K**) ER-Golgi connections. (**A**,**B**) Serial tomography slices show bead-like connections (red arrows) between the ER and the Golgi cisterna within the crater. (**C**–**E**) The connection is shown from different sides of view. Green color indicates the ER. (**F**,**G**) Red smooth spheres are the 52-nm vesicles. Red structures with rough surface are COPI coated buds. (**F**,**G**) The connection (red arrows) shown within the entire 3D model without elimination of cisternae from opposite side of view. (**G**) The purple color shows another example of the bead-like connection (red arrows). (**H**,**K**) The plasticine model of these connections (red arrows) within the crater (green arrows). (**I**,**J**) Examples of connections (red arrows) between the ER (green color) and Golgi compartments (violet to the left and yellow to the right). (**L**–**N**) Serial tomo-images of the connection between the ER and the Golgi existing within the rim of the Golgi cup edge. Red arrows show its way from one membrane to another. (**O**,**P**) Fusion of SGs (purple arrows). Green arrows indicate the remnants of membranes inside newly formed large SG. Scale bars (nm): 75 nm (**A**,**B**); 80 (**C**,**D**); 60 (**E**); 70 (**F**,**G**,**L**–**N**); 50 (**I**,**J**); 1000 ((**O**); below the image); 280 (**P**).

**Figure 4 cells-14-01185-f004:**
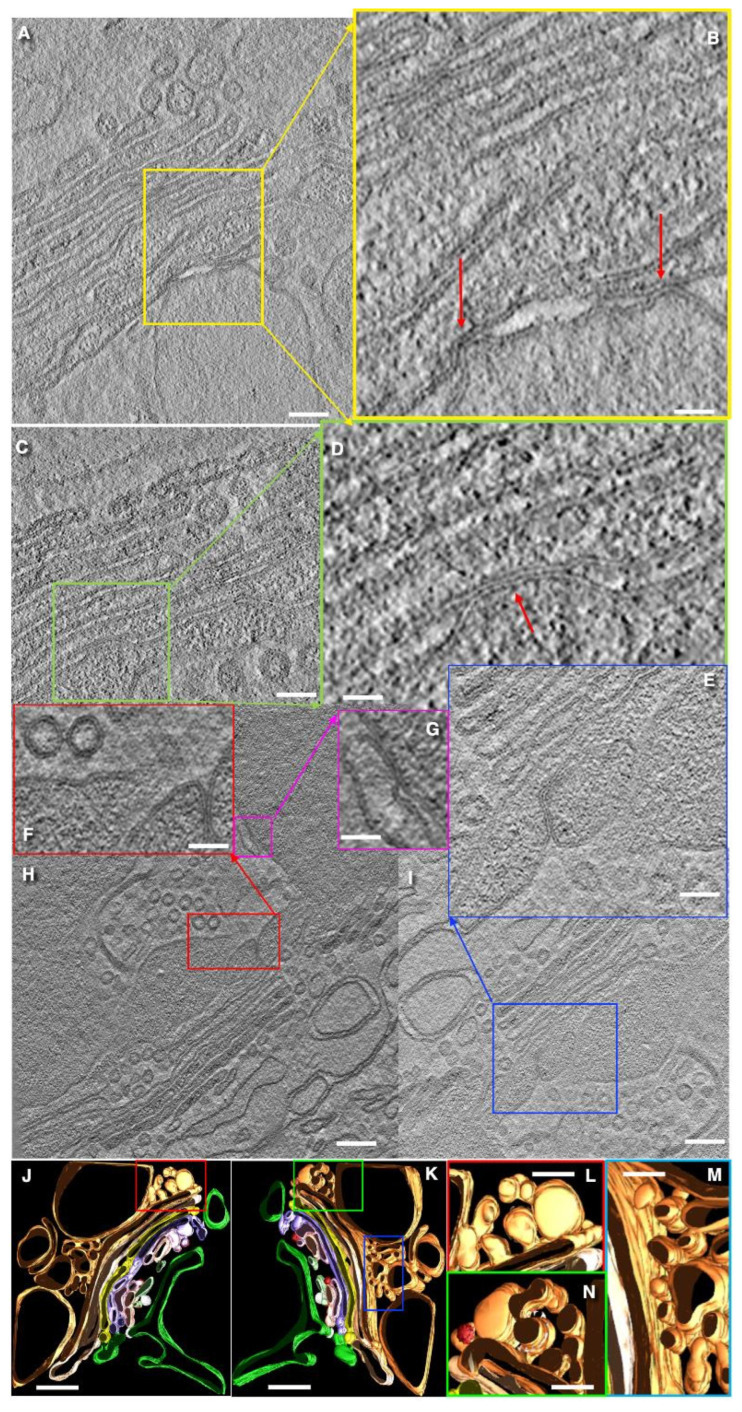
Close contacts (red arrows) between cisternae and between secretory granules and cisternae. (**A**–**I**) Membranes of Golgi cisternae and SG are tightly attached to each other and form a contact composed of five layers (or three dark layers). (**J**–**M**) The 3D reconstruction of the Golgi ribbon in goblet cells. View from different sides. Green color indicates the ER. Here, we have two seemingly similar but not the same images. The image (**H**) and image (**I**) are serial images take from two tomograms obtained after rotation of the same sample in the microscope of 90 degrees in order to obtain the double tilt tomogram. However, this peculiar contact structure composed of three dark lines was more visible in these separated tomograms, and these serial images were used for the illustration. (**L**–**M**) Enlarged areas inside the red, green, and blue boxes in (**J**,**K**). The panel (**N**) represents a full image. In (**N**), we demonstrated only the parts (in boxes indicated by green and red boxes) of the whole images (see (**J**,**K**)). The whole figures are shown in (**J**,**K**). Scale bars (nm): 92 (**A**); 30 (**B**,**G**); 80 (**C**,**E**,**L**–**N**); 25 (**D**); 75 (**F**); 160 (**H**,**I**); 260 (**J**,**K**).

**Figure 5 cells-14-01185-f005:**
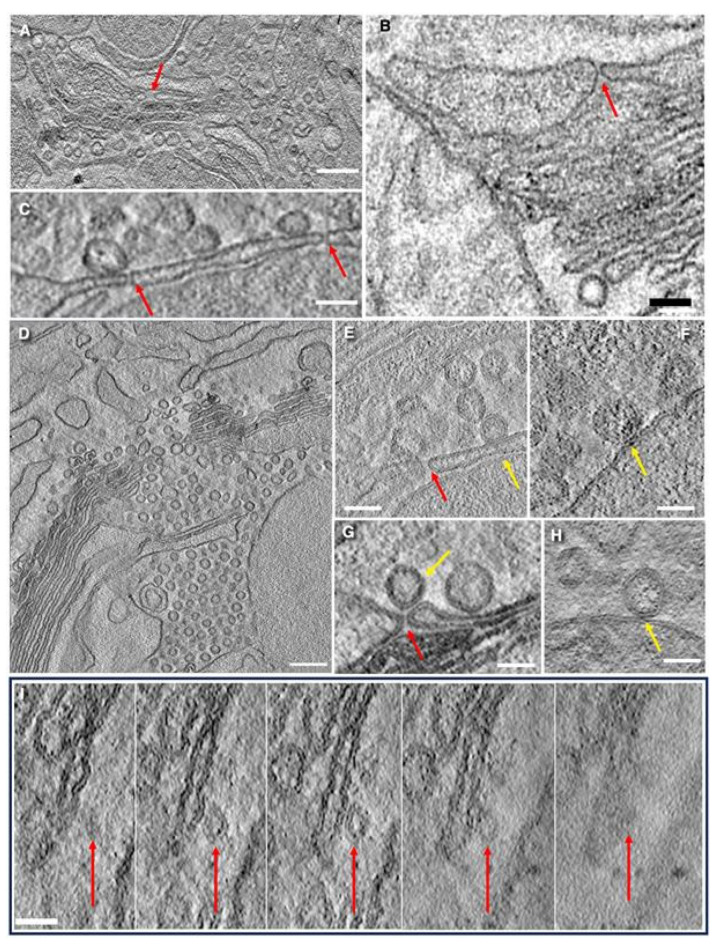
Structure of the Golgi in goblet cells. (**A**–**C**,**E**,**G**) Pores (red arrows) near the boundary cisternal distensions and the 52 nm vesicles (yellow arrows in (**E**–**H**)) attached to membrane of this cisterna. (**B**) The red arrow shows the pore separating the boundary cisternal distension from the rest of the cisterna. (**D**) Cluster of the 52 nm presumably COPI-dependent vesicles. (**E**–**H**) The 52 nm vesicles attached to membrane of Golgi cisternae (yellow arrows). (**I**) Serial EM tomography images demonstrating the 42 nm vesicle. Red arrows show the site where the 42 nm vesicles appear and then disappear. Scale bars (nm): 200 (**A**,**D**); 70 nm (**B**); 50 (**C**,**F**–**H**); 75 (**E**,**I**).

**Figure 6 cells-14-01185-f006:**
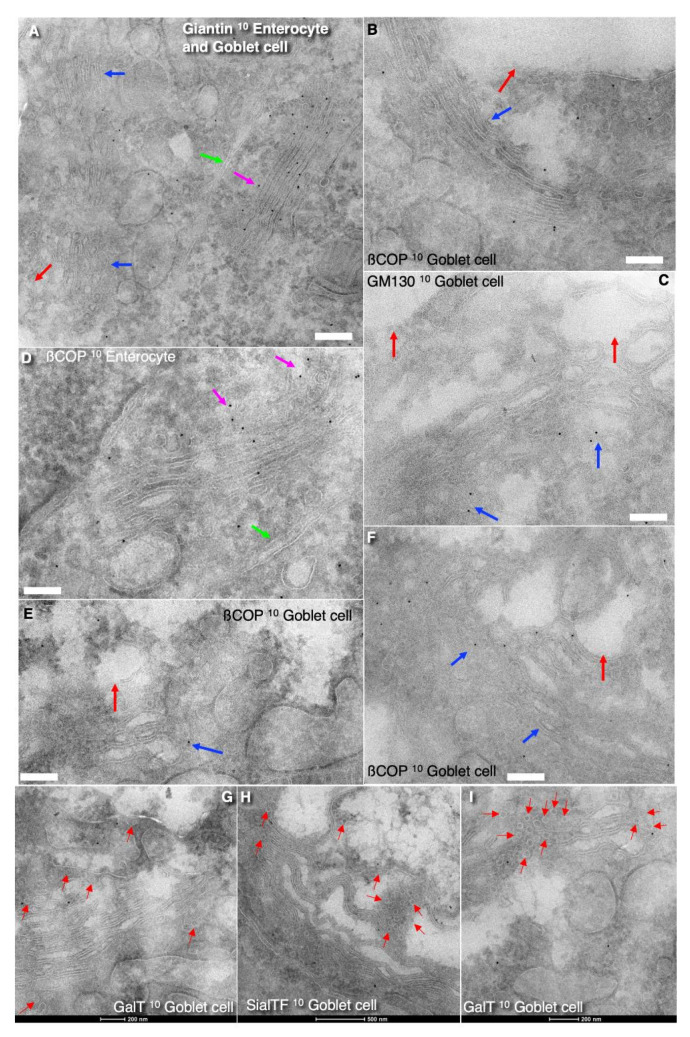
Immune EM labeling of the Golgi in goblet cells and neighboring enterocytes for Golgi markers. Red arrows indicate secretory granules. Blue arrows show the Golgi of goblet cells. Magenta arrows demonstrate the Golgi of enterocytes. Green arrows show the PM immune EM labeling for routinely used Golgi enzymes, namely GalT (**G**,**I**) and SialTF (**H**). No gold over the 52 nm vesicles was visible. Labeling of the Golgi in goblet cells and neighboring enterocytes for giantin (**A**), ß-COP (**B**,**D**–**F**), and GM130 (**C**). Expression of these markers in the Golgi of goblet cells is lower than in the Golgi of absorptive enterocytes. Scale bars (nm): 285 (**A**–**C**,**E**,**F**); 192 (**D**). In (**G**–**I**) scale bars are shown in the bottom of images.

**Figure 7 cells-14-01185-f007:**
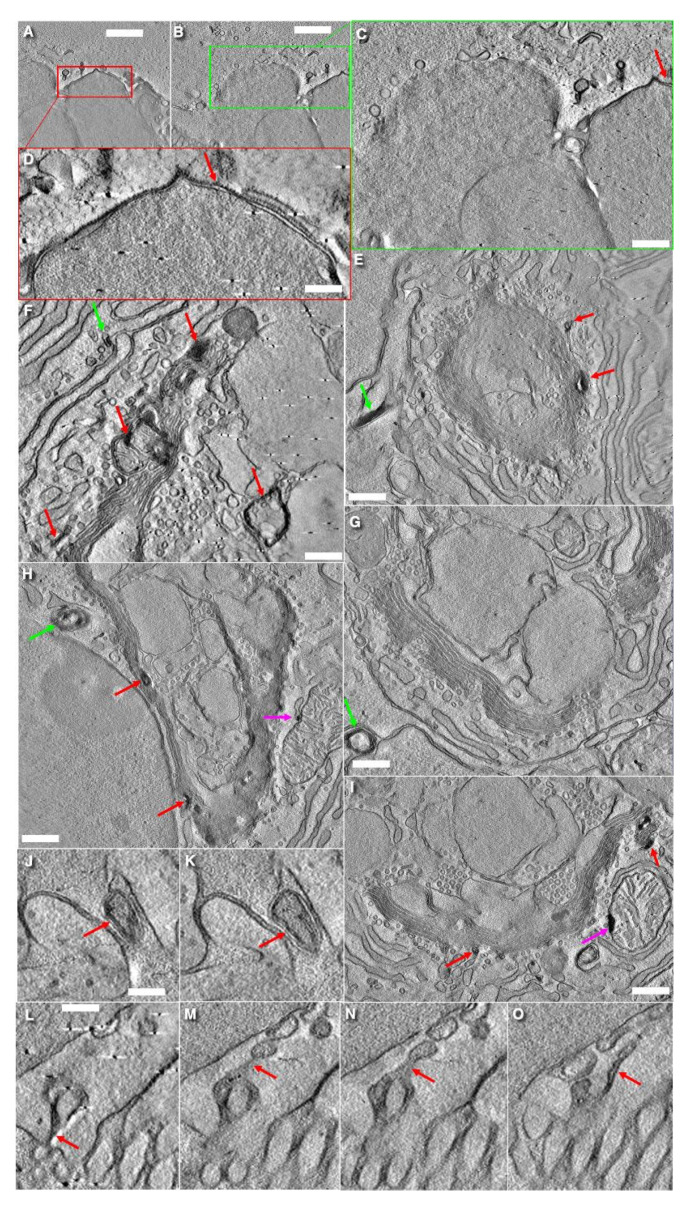
Possible mechanisms involved in elimination of the excess membranes within APM after the fusion of subapical SG with the APM. (**A**,**B**) Apical part of goblet cells exhibits formation of a bubble filled with the SG and its extrusion into the intestinal lumen. (**C**) Enlargements of the inside of the green box in (**B**). (**D**) Enlargement of the area inside the red box in (**A**). Double membranes are shown with red arrows. Dark dots on the surface of the APM represent the molecules of Muc1. (**E**–**I**) Accumulation of MLOs within the ER, ERES, and the Golgi and, subsequently, their secretion (**E**,**G**,**H**) into the intercellular space. MLOs within the cis side of the Golgi mimic the first three-osmiophilic cisternae (**H**). Green arrows show MLOs in the space between epithelial cells. Red arrows indicate MLOs within the Golgi. Magenta arrows demonstrate MLOs connected with the OMM (**H**,**I**). (**J**,**K**) Serial images of MLO (red arrows) near the APM. (**L**–**O**) Serial tomo-images of the apical part of a goblet cell near the villous enterocytes (to the right). For the full images at low magnifications, see Appendix A. Scale bars (nm): 855 (**A**,**B**); 190 (**C**,**D**); 385 (**E**); 300 (**F**); 750 (**G**–**I**); 240 (**J**–**O**). In (**L**–**O**) bar is the same.

**Figure 8 cells-14-01185-f008:**
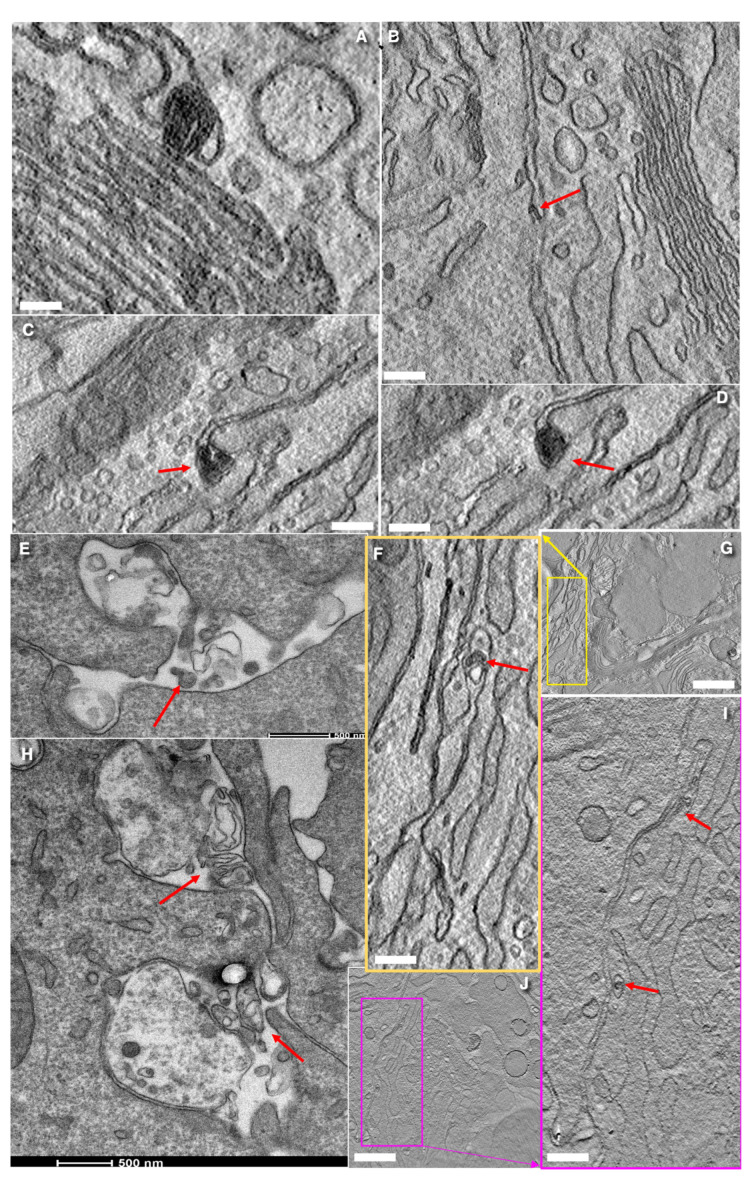
Accumulation of MLOs (red arrows) and their movement to the ER, into the Golgi, and into the intercellular space observed in quiet goblet cells presumably after massive secretion. (**A**) MLO (red arrow) situated between the ER and the Golgi. (**B**,**F**,**G**,**I**,**J**) MLO (red arrows) in the space between cells. (**C**,**D**) Serial tomographic images of MLO (red arrows) in the space between goblet cell and enterocytes. (**E**,**H**) Secretion of MLOs (red arrows) into intercellular space. Scale bars (nm): 260 (**A**); 142 (**B**–**D**,**E**); 500 (**F**–**H**,**I**); 780 (**J**). In images (**E**,**H**) the bars and their size are visible below the images.

**Table 1 cells-14-01185-t001:** Morphometric parameters of goblet cells in the mouse colon.

Parameters ± SD (Number of Samples Is Equal to 6)		Colon	
		Rest.	Secr.
A. The percentage of goblet cells that have contacts with dendritic cells or their processes.		nd	61 ± 7
B. The mean number of Golgi cisternae in stacks		7.7 ± 2.4	7.1 ± 2.1
C. The mean volume of the Golgi ribbon (µm^3^)		nd	49.1 ± 10
D. The mean surface area of Golgi cisternae (µm^2^)		nd	1390 ± 134
E. Average diameter of secretory granules (µm)		1.2 ± 0.2 *	2.5 ± 0.31 *
F. The ratio of the surface areas of the rims of the cisternae to the area of their flat zones in different cisternal layers	First layer	0.33 ± 0.03	nd
	Second layers	0.31 ± 0.09	nd
	Third layer	0.27 ± 0.07	nd
	Fourth layer	0.20 ± 0.05	nd

* The difference between two means is statistically significant (*p* < 0.05).

**Table 2 cells-14-01185-t002:** Labeling density of different markers of goblet cells in the mouse colon.

A. The ratio (%) between the LD for LC3 and Muc1 over MLOs versus background	LC3	Muc1
	4.1 ± 0.9 *	4.6 ± 1.1 *
B. The ratio (%) between the LD for Golgi enzymes (GalT and (SialTF) over the 52 nm vesicles versus the plane parts of cisternae	GalT	SialTF
	32 ± 14 **	41 ± 17 **
C. The ratio of the labeling density (LD) for ßCOP over the Golgi of neighboring enterocytes with microvilli to that of goblet cells	nd	4.9 ± 1.5 ^
D. The ratio of LD for giantin in the Golgi of the neighboring enterocytes with microvilli to the same parameter in goblet cells	nd	5.2 ± 1.4 ^^

* The difference between the mean and background is statistically significant (*p* < 0.05). ** The difference between the mean and background is statistically significant (*p* < 0.05). ^ The ratio between LD over the 52 nm vesicles and cisternae is statistically significant (*p* < 0.05). ^^ The ratio between LD over the 52 nm vesicles and cisternae is statistically significant (*p* < 0.05).

## Data Availability

The original contributions presented in this study are included in the article/Appendix A. Further inquiries can be directed to the corresponding author.

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
