# Peer review of "Structure of the Secretory Compartments in Goblet Cells in the Colon and Small Intestine"

_cells, 2025, doi:10.3390/cells14151185_

Round 1
Reviewer 1 Report
Comments and Suggestions for Authors
The organization of the secretary pathway in goblet cells remains poorly understood in terms of the compartmental boundaries and in the later stages of the mucins release from large granules. The authors are experts electron microscopists and have made significant contributions to the overall process of protein secretion. The images are impressive, but I am surprised to see the lack of quantitation and the use of immunoelectron microscopy to describe compartments of the secretory pathway. The authors spend too much space in describing the events in mucins maturation by glucoyslation in the Golgi, and then in the events leading to their condensation post packaging into micrometer size granules. There is also a description of the ER exit sites, but there is no quantitation in terms of their numbers or the overall morphology.
Also, are these images of resting goblet cells or activated goblet cells? The latter produce mucins and release them in a signal dependent manner. The organization of the secretory pathway therefore would be different based on the state of the goblet cells.
Does blocking export from the Golgi or fusion of the mucin granules changes the organization of the secretory pathway? Simple chemical inhibitors can be used to gain valuable insights.
Author Response
Reply
Trying to fulfil all demands of our excellent reviewers, we completely re-wrote the manuscript and replaced most Figures. We will correct the reference format after final decision on the paper.
Reviewer 1
- The organization of the secretary pathway in goblet cells remains poorly understood in terms of the compartmental boundaries and in the later stages of the mucins release from large granules. The authors are experts electron microscopists and have made significant contributions to the overall process of protein secretion. The images are impressive, but I am surprised to see the lack of quantitation and the use of immunoelectron microscopy to describe compartments of the secretory pathway.
Our reply
Thanks a lot for your excellent work. We added some new data on immune labelling and quantitation. However, the detailed analysis of the Golgi labelling. will be presented in the next paper. Additionally we added the mathematical models of the Golgi structure.
- The authors spend too much space in describing the events in mucins maturation by glycosylation in the Golgi, and then in the events leading to their condensation post packaging into micrometre size granules.
Our reply
We made these parts of the text shorter.
- There is also a description of the ER exit sites, but there is no quantitation in terms of their numbers or the overall morphology.
Our reply
We added quantitative data
- Also, are these images of resting goblet cells or activated goblet cells? The latter produce mucins and release them in a signal dependent manner. The organization of the secretory pathway therefore would be different based on the state of the goblet cells.
Our reply
The organization of the secretory pathway in Goblet cells with open apical plasmalemma and closed one differs only in the state of secretory granules. The difference is related to the structures of secretory granules near the apical plasmalemma or rarely of secretory granules situated deeper but not the Golgi per se. The Golgi has almost the same structure in both kinds of cells. We indicated this in the text.
- Does blocking export from the Golgi or fusion of the mucin granules changes the organization of the secretory pathway? Simple chemical inhibitors can be used to gain valuable insights.
Our reply
In organs and tissues, synchronization of the proximal parts of the secretory pathway is almost impossible. Under these conditions, namely, the opened apical plasmalemma and closed one, the organization of the secretory pathway, especially the Golgi is almost the same. The secretion affects only the very distal parts of the pathway. The single change is the fusion of secretory granules and secretion of mucins from the secretory granule situated near the APM.

Reviewer 2 Report
Comments and Suggestions for Authors
Cells-3536479
Structure of the Golgi in Goblet cells
Goblet cells are specialized for the secretion of mucins, highly O-glycosylated proteins with high molecular weights. In the present study, the authors analyzed the structure of the Golgi complex (GC) of goblet cells using high-resolution three-dimensional electron microscopy, and found that GC are composed of 7 cisternae, which form cup-like spiral structures filled with secretory granules containing mucins. The authors also suggest that the intracellular transport of mucins is explained by the kiss-and-run model combined with the diffusional model. This manuscript is of potential interest, however, it is very difficult to follow their results and discussions as detailed below.
- Some of the figure panels in this manuscript seem completely the same as those published in their previous manuscript: Ref. 27, IJMS 2023, except some are rotated at 90 or 180 degrees. Here is the list of reused figures that this reviewer found by eyes:
Fig. 1C
Fig. 2A, B
Fig. 3F, G, H, I, K
Fig.4A, B, E, F, G, H, I, K
Fig. 6A
Fig. S1 A, B, C, D, F
Reuse of the figures is not cited in the text or in the figure legends.
- Since figure legends are not described appropriately and contains many errors, it is very difficult to understand what the authors want to claim based on their results. In addition, some legends explain another figures.
Also, in the result section, each figure is not explained, and at the end of the sentences, all the panels in one figure are cited; e. g., Fig. 2A-E (lines 279 and 295), Fig3, 4A-K (line 347), and so on.
- Number of the samples analyzed are not described; e.g., 52+5% Goblet cells (line 262), 58+7% ER-Golgi connections are (line 363) and so on.
- Scale bars are missing in some figures.
- They claim that “ER cisternae contain small ER exit sites (ERES)” (lines 337-338). However, no data on which they deduced this conclusion is shown. It is not clear how the authors identified the ERES.
Similarly, the authors describe on bead-shaped tubule (line 341), but the data or figure showing this tubule is not cited, thus it is difficult to understand their claim.
Also, the data showing “ERES are located near the craters” (lines 341-342) is not clear.
Importantly, the reason why the authors suggest the kiss-and-run model for the intracellular transport of mucins are not appropriately explained.
- Introduction is too long, and contains a detailed description on the identification of mucins by lectins, which can be removed safely.
- Lines 541-546: the sentences are short like notes.
Comments on the Quality of English Language
There are many grammatical errors, typos, and too short sentences.
Author Response
Reviewer 2
- Goblet cells are specialized for the secretion of mucins, highly O-glycosylated proteins with high molecular weights. In the present study, the authors analysed the structure of the Golgi complex (GC) of goblet cells using high-resolution three-dimensional electron microscopy, and found that GC are composed of 7 cisternae, which form cup-like spiral structures filled with secretory granules containing mucins. The authors also suggest that the intracellular transport of mucins is explained by the kiss-and-run model combined with the diffusional model. This manuscript is of potential interest; however, it is very difficult to follow their results and discussions as detailed below.
Our reply
Thanks a lot for your excellent work. We made our text simpler.
- Some of the figure panels in this manuscript seem completely the same as those published in their previous manuscript: Ref. 27, IJMS 2023, except some are rotated at 90 or 180 degrees. Here is the list of reused figures that this reviewer found by eyes:
Fig. 1C
Fig. 2A, B
Fig. 3F, G, H, I, K
Fig.4A, B, E, F, G, H, I, K
Fig. 6A
Fig. S1 A, B, C, D, F
Reuse of the figures is not cited in the text or in the figure legends.
Our reply
We replaced these figures with similar new ones.
- 3. Since figure legends are not described appropriately and contains many errors, it is very difficult to understand what the authors want to claim based on their results. In addition, some legends explain another figures.
Our reply
We re-wrote the text.
- 4. Also, in the result section, each figure is not explained, and at the end of the sentences, all the panels in one figure are cited; e. g., Fig. 2A-E (lines 279 and 295), Fig3, 4A-K (line 347), and so on.
Our reply
We corrected these mistakes.
- Number of the samples analysed are not described; e.g., 52+5% Goblet cells (line 262), 58+7% ER-Golgi connections are (line 363) and so on.
Our reply
We included these data.
- Scale bars are missing in some figures.
Our reply
We included bars.
- They claim that “ER cisternae contain small ER exit sites (ERES)” (lines 337-338). However, no data on which they deduced this conclusion is shown. It is not clear how the authors identified the ERES.
Our reply
ERES were identified on the basis of the criteria proposed by Bannykh et al. (1996). We cannot compare the size of ERES in Goblet cells with that in absorptive enterocytes and compared with thar described by Bannykh et al. (1996) and Mironov et al. (2003).
- Similarly, the authors describe on bead-shaped tubule (line 341), but the data or figure showing this tubule is not cited, thus it is difficult to understand their claim.
Our reply
We made additional membrane rendering and tomography analysis and not this phenomenon is shown better.
- Also, the data showing “ERES are located near the craters” (lines 341-342) is not clear.
Our reply
New images and descriptions were added.
- Importantly, the reason why the authors suggest the kiss-and-run model for the intracellular transport of mucins are not appropriately explained.
Our reply
We made the text clearer
- Introduction is too long, and contains a detailed description on the identification of mucins by lectins, which can be removed safely.
Our reply
We made Introduction shorter and eliminated the part on mucins.
- Lines 541-546: the sentences are short like notes.
Our reply
We corrected this.
- There are many grammatical errors, typos, and too short sentences.
Our reply
We corrected all mistakes.

Round 2
Reviewer 1 Report
Comments and Suggestions for Authors
The written English is better, but it still suffers from the same problem as before. The authors make amazing assumption with clear data. Here’s one example,
Figure 3: Where is the evidence that the proposed connections are between ER and the Golgi cisternae?
One more issue that the reviewers might want to consider is that direct connection between ER exit sites and the cis Golgi cisternae has been shown by Jose Pastor and colleagues in fly cells.
Author Response
Reviewer 1
- The written English is better, but it still suffers from the same problem as before. The authors make amazing assumption with clear data. Here’s one example Figure 3: Where is the evidence that the proposed connections are between ER and the Golgi cisternae?
Our reply
Thanks a lot. We corrected this figure and presented this and additional better visible ER-Golgi connections.
- One more issue that the reviewers might want to consider is that direct connection between ER exit sites and the cis Golgi cisternae has been shown by Jose Pastor and colleagues in fly cells.
Our reply
Thanks a lot. In the new variant of the manuscript, we quoted this paper (Liu M, Feng Z, Ke H, Liu Y, Sun T, Dai J, Cui W, Pastor-Pareja JC. 2017. Tango1 spatially organizes ER exit sites to control ER export. J Cell Biol. 2017. 216(4):1035-1049. doi: 10.1083/jcb.201611088.). However, in this paper, there is no confirmation of ER-Golgi connecting using three-dimensional electron microscopy or FRAP analysis. Moreover, we quoted data demonstrating ER-Golgi connections several times. (Mironov et al., 1997; Mironov et al. 2003; Mironov et al., 2005; Cutrona et al. 2013; Mironov and Beznoussenko, 2019). In these papers, our reviewer could see dozens of such data demonstrating ER-Golgi connections. Thus, ER-Golgi connections is normally existing structures. However, in majority of cases this structure is not constant.

Reviewer 2 Report
Comments and Suggestions for Authors
Cells-3536479: Revise
Structure of the Golgi in goblet cells supports the diffusion model of intra-Golgi transport there
The manuscript has been changed in large part, but still, hard to understand the authors claims as detailed below.
Major points:
- The rationale for the manuscript organization is not clear. The conclusion of this manuscript is the diffusion model of the intra-Golgi transport of mucin in goblet cells, however, in the last figure (Fig. 7), the authors describe the formation and function of MLOs. What is the significance of the MLOs in the context of diffusion model of intra-Golgi transport?
- The abstract does not represent correctly what they have shown in this manuscript.
- What is the significance of reduced number of GM130 and betaCOP in goblet cells (Fig. 6)? Describe the interpretation of these results.
- Still many errors in the figures and fig. legends, which is confusing to the reader:
- 1, legends are inconsistent in lines 406-407 and lines 412-413.
- 2, green arrows (line 431) and blue arrows (line 433) are missing in the Fig.
- 3, red arrow (line 493) is missing in the Fig.
- 4, Panel H and I are the same, except they are rotated 90 degrees, each other. What does this mean???
Only a part of panel N is shown; Fig. legend on panel N is missing.
important part of panel D is masked by panel E.
- 6, Label in Panel A is missing.
- 7, line 855: maybe incorrect.
- Provide an overview of what is known so far about the diffusion model in the Introduction. Also, explain how the authors conclude the intra-Golgi transport by diffusion in the result or in discussion.
- The result for Fig. 4 should come before that of Fig. 5.
- The nature of 42-nm vesicles is not clearly described.
- Lines 437-438: Explain how COPI and COPII coated buds are identified.
- Mucin precipitations (lines 608, 610, 615) are not visible in the EM figures.
- What does the structure composed of five membrane layers mean? (lines 706-707, and 834)
- Mucin precipitation are not visible (lines 608, 610, 615) in the EM figures.
- Two different “Fig. S1” are attached.
Minor points:
- Many typos and grammatical errors need to be corrected.
- Describe the antibodies used for the immunoEM.
Many typos and grammatical errors need to be corrected.
Author Response
Reviewer 2
Cells-3536479: Revise
Structure of the Golgi in goblet cells supports the diffusion model of intra-Golgi transport there. The manuscript has been changed in large part, but still, hard to understand the authors claims as detailed below.
We thank our reviewer for a huge and very useful work.
Major points:
- The rationale for the manuscript organization is not clear. The conclusion of this manuscript is the diffusion model of the intra-Golgi transport of mucin in goblet cells, however, in the last figure (Fig. 7), the authors describe the formation and function of MLOs. What is the significance of the MLOs in the context of diffusion model of intra-Golgi transport?
Our reply
We corrected the text, rewrote the abstract, and included in the Introduction the explanation of the necessity to apply new knowledge for the analysis of several function of goblet cells. We added results of a new set of immune EM analysis, demonstrating that the 52-nm vesicles have lower concentration of Golgi enzymes than Golgi cisternae, and that MLOs are positive for LC3 and mucin-2.
Also, the presence of dendritic cells around goblet cells suggests that there are some kind of process related to immunity. Moreover it is unclear how the excessive apical membrane formed after fusion of secretory granule with the apical plasma membrane is eliminated. We thought that this process is related to the function of the Golgi. However in order to eliminate the irritative element related to models of intracellular transport which affect the vesicular and maturation dogmas we returned our first title.
Additionally we included in the Introduction the explanation of the necessity for our analysis was directed to understand why manipulation with the same miRNA gave different results tin these cells (Briata et al., 2021). The low level of expression of giantin, COPI and GM130 in the goblet Golgi than in the Golgi of villous enterocytes explains why manipulation with the same miRNA gave different results tin these cells. Finally, considering the full secretion process we could avoid the explanation of the following issue. After fusion of membrane of secretory granules with apical plasmalemma the surface area of the latter increased greatly. The mechanism of elimination of this membrane was not known.
Our analysis of goblet cells allowed us to identify three types of cells. The first is with homogeneous secretory granules that do not coalesce on a large scale. The second group included cells that were in the process of secretion and had APM tears connecting the intestinal lumen to the lumen of the emptying secretory granule. The third type of cells, visible less frequently had a complete APM, but signs of fusion of secretory granules. In the apical part of such cells, we found MLOs connected by a double membrane to the APM. The second option for removing excess APM could be the formation of a protrusion of a secretory granule inserted into the APM membrane, from which mucus was removed, into the intestinal lumen and the inclusion of another secretory granule there, which, being surrounded by a double membrane, split off into the intestinal lumen. We made several EM tomograms on the apical zone of goblet cells and added Figure 8. This information was added into the text.
- The abstract does not represent correctly what they have shown in this manuscript.
Our reply
We rewrote the abstract.
- What is the significance of reduced number of GM130 and beta-COP in goblet cells (Fig. 6)? Describe the interpretation of these results.
Our reply
We included this explanation.
- Still many errors in the figures and fig. legends, which is confusing to the reader:
Our reply
We corrected the mistakes
- Legends are inconsistent in lines 406-407 and lines 412-413.
Our reply
We changed the text there.
- Green arrows (line 431) and blue arrows (line 433) are missing in the Fig.
Our reply
We corrected this figure.
- Red arrow (line 493) is missing in the Fig.
Our reply
We corrected this figure.
- Panel H and I are the same, except they are rotated 90 degrees, each other. What does this mean???
Our reply
The single images suitable for this description is Figures 4H , I. Indeed, here we have two similar but not the same images. The imаge H and image I are serial images take from two tomograms obtained after rotation of the same sample in the microscope on 90 degree in order to get double tilt tomogram. However, this peculiar contact structure composed of three dark lines was visible better in separated tomograms and used these serial images for the illustration. We added additional explanation in the legend.
- Only a part of panel N is shown; Fig. legend on panel N is missing.
Our reply
Panels N exist in Figures 1 and 4. In Figure 1, the panel N is shown as a full images. In Figure 4N, we demonstrated only the parts (in boxes indicated by green and red boxes) of the whole images (see Figure 4J, K). The whole figure could be examined in Figures 4J, K.
- Important part of panel D is masked by panel E.
Our reply
We corrected the image.
- Label in Panel A is missing.
Our reply.
We corrected the images and added several additional images demonstrating that concentration of Golgi enzymes in the 52-nm vesicles is lower than in cisternae.
- Line 855: maybe incorrect.
Our reply
We eliminated this mistake.
- Provide an overview of what is known so far about the diffusion model in the Introduction. Also, explain how the authors conclude the intra-Golgi transport by diffusion in the result or in discussion.
Our reply
In the discussion, we added quotation of our reviews and presented a very short description of the model and explanation why we made such kind of conclusion.
- The result for Fig. 4 should come before that of Fig. 5.
Our reply
We corrected this mistake.
- The nature of 42-nm vesicles is not clearly described.
Our replay
Until now, this nature is not known. We were the first in presentation of images of the 42-nm vesicles in absorptive enterocytes (Sesorova et al,. 2020). We submitted the paper in "Cells" where we analyse the diameters of cells vesicles and described these vesicles as clathrin-coated and clathrin-dependent similar to those in neuronal body. There, the 42-nm vesicles are transported to synapses and become synaptic vesicles.
- Lines 437-438: Explain how COPI and COPII coated buds are identified.
Our reply
We submitted the paper into Cells where this is explained. Briefly: COPII-coated buds have diameter equal to 70-80 nm and their coat is 1.1-fold thicker. COPI-coated buds have diameter equal to 50-55 nm and their coat is more osmiophilic than the COPII-coat. We added this explanation to Methods.
- Mucin precipitations (lines 608, 610, 615) are not visible in the EM figures.
Our reply
Mucin precipitation cannot be visualized in electron microscope without additional staining. However, our attempts to use known dyes were unsuccessful. We conclude that mucins are subjected to precipitation on the basis of the widening of the lumen of Golgi cisternae.
- What does the structure composed of five membrane layers mean? (lines 706-707, and 834)?
Our reply.
Application of OsO4 for fixation induces formation of three lines on membranes, namely, external and internal electron dense lines represent precipitation of Os along lipid heads. Fatty acids between these lone are less dense due to lower level of Os precipitation (Bozzola, J.J.; Russell, L.D. 1999. Electron Microscopy: Principles and Techniques for Biologists. Jones and Bartlett, Boston, 670 p.).
- Mucin precipitation are not visible (lines 608, 610, 615) in the EM figures.
Our reply
Mucin precipitation cannot be visualized in electron microscope without additional staining. However, our attempts to use known dyes were unsuccessful. We conclude that mucins are subjected to precipitation on the basis of the widening of the lumen of Golgi cisternae.
- Two different “Fig. S1” are attached.
Our reply
We corrected this mistake.
- Minor points: 1. Many typos and grammatical errors need to be corrected.
Our reply
We corrected mistakes.
- Describe the antibodies used for the immuno-EM.
Our reply
We described these antibodies in the text and in Figure 4A.

Round 3
Reviewer 1 Report
Comments and Suggestions for Authors
Publish
Author Response
Reply 14 07 25
Reviewer 1 (third round)
Open Review
(x) I would not like to sign my review report
( ) I would like to sign my review report
Quality of English Language
( ) The English could be improved to more clearly express the research.
(x) The English is fine and does not require any improvement.
|
Yes |
Can be improved |
Must be improved |
Not applicable |
|
|
Does the introduction provide sufficient background and include all relevant references? |
(x) |
( ) |
( ) |
( ) |
|
Is the research design appropriate? |
(x) |
( ) |
( ) |
( ) |
|
Are the methods adequately described? |
(x) |
( ) |
( ) |
( ) |
|
Are the results clearly presented? |
(x) |
( ) |
( ) |
( ) |
|
Are the conclusions supported by the results? |
( ) |
( ) |
( ) |
( ) |
|
Are all figures and tables clear and well-presented? |
(x) |
( ) |
( ) |
( ) |
Comments and Suggestions for Authors
Publish
Our reply: Thanks a lot.
Reviewer 2 Report
Comments and Suggestions for Authors
Cells-3536479: Revise2
Structure of the secretory compartments in goblet cells in colon and small intestine
The manuscript still needs to be improved.
- To improve the readability, results should contain several headings.
- Discussion is too long. Discuss on the results presented in the manuscript.
- Figures and figure legends still contain the similar problems as pointed previously.
- Diffusion model is now well explained, but still, it is not clear where the kiss and run model works.
- Discussion on glucose-containing polysaccharides as the reason of osmiophilic cisternae is not clear (lines 567-587).
- Still, many typos and grammatical errors remain to be corrected.
Author Response
Reviewer 2 (third round)
The manuscript still needs to be improved.
- To improve the readability, results should contain several headings.
Our reply: We did this.
- Discussion is too long. Discuss on the results presented in the manuscript.
Our reply: We made this chapter shorter.
- Figures and figure legends still contain the similar problems as pointed previously.
Our reply: We checked this problem again and improved these figures and legends. Also, we added several supplementary figures from our previous papers explaining different models of intracellular transport.
Here we place our previous reply on the comment presented in the second round.
Reviewer 2: previous comments on Figures from second round
- Still many errors in the figures and fig. legends, which is confusing to the reader:
Our reply
We corrected the mistakes
- Legends are inconsistent in lines 406-407 and lines 412-413.
Our reply
We changed the text there.
- Green arrows (line 431) and blue arrows (line 433) are missing in the Fig.
Our reply
We corrected this figure.
- Red arrow (line 493) is missing in the Fig.
Our reply
We corrected this figure.
- Panel H and I are the same, except they are rotated 90 degrees, each other. What does this mean???
Our reply
The single images suitable for this description is Figures 4H , I. Indeed, here we have two similar but not the same images. The imаge H and image I are serial images take from two tomograms obtained after rotation of the same sample in the microscope on 90 degree in order to get double tilt tomogram. However, this peculiar contact structure composed of three dark lines was visible better in separated tomograms and used these serial images for the illustration. We added additional explanation in the legend.
- Only a part of panel N is shown; Fig. legend on panel N is missing.
Our reply
Panels N exist in Figures 1 and 4. In Figure 1, the panel N is shown as a full images. In Figure 4N, we demonstrated only the parts (in boxes indicated by green and red boxes) of the whole images (see Figure 4J, K). The whole figure could be examined in Figures 4J, K.
- Important part of panel D is masked by panel E.
Our reply
We corrected the image.
- Label in Panel A is missing.
Our reply.
We corrected the images and added several additional images demonstrating that concentration of Golgi enzymes in the 52-nm vesicles is lower than in cisternae.
- Line 855: maybe incorrect.
Our reply
We eliminated this mistake.
- Provide an overview of what is known so far about the diffusion model in the Introduction. Also, explain how the authors conclude the intra-Golgi transport by diffusion in the result or in discussion.
Our reply
In the discussion, we added quotation of our reviews and presented a very short description of the model and explanation why we made such kind of conclusion.
- The result for Fig. 4 should come before that of Fig. 5.
Our reply
We corrected this mistake.
- Mucin precipitations (lines 608, 610, 615) are not visible in the EM figures.
Our reply
Mucin precipitation cannot be visualized in electron microscope without additional staining. However, our attempts to use known dyes were unsuccessful. We conclude that mucins are subjected to precipitation on the basis of the widening of the lumen of Golgi cisternae.
- What does the structure composed of five membrane layers mean? (lines 706-707, and 834)?
Our reply.
Application of OsO4 for fixation induces formation of three lines on membranes, namely, external and internal electron dense lines represent precipitation of Os along lipid heads. Fatty acids between these lone are less dense due to lower level of Os precipitation (Bozzola, J.J.; Russell, L.D. 1999. Electron Microscopy: Principles and Techniques for Biologists. Jones and Bartlett, Boston, 670 p.).
- Mucin precipitation are not visible (lines 608, 610, 615) in the EM figures.
Our reply
Mucin precipitation cannot be visualized in electron microscope without additional staining. However, our attempts to use known dyes were unsuccessful. We conclude that mucins are subjected to precipitation on the basis of the widening of the lumen of Golgi cisternae.
- Two different “Fig. S1” are attached.
Our reply
We corrected this mistake.
- Diffusion model is now well explained, but still, it is not clear where the kiss and run model works.
Our reply: We included supplementary figures and small part of the text with additional explanation.
- Discussion on glucose-containing polysaccharides as the reason of osmiophilic cisternae is not clear (lines 567-587).
Our reply: We made the text simple and quote Bozzola and Russel 1992).
- Still, many typos and grammatical errors remain to be corrected.
Our reply: We checked the text once more.